# Atmospheric conditions favouring extreme precipitation and flash floods in temperate regions of Europe

Judith Meyer[1,2], Malte Neuper[3], Luca Mathias[4], Erwin Zehe[3], Laurent Pfister[1,2]

[1] Catchment and Ecohydrology Group (CAT), Environmental Research and Innovation, Luxembourg Institute of Science and
5 Technology (LIST), Belvaux, 4422, Luxembourg
[2] Faculty of Science, Technology and Medicine (FSTM), University of Luxembourg, Esch-sur-Alzette, 4365, Luxembourg
[3] Institute of Water Resources and River Basin Management, Karlsruhe Institute of Technology (KIT), Karlsruhe, Germany
[4] Air Navigation Administration, MeteoLux, Findel, Luxembourg

10 *Correspondence to*: Judith Meyer (judith.meyer@list.lu)

**Abstract:** In recent years, flash floods repeatedly occurred in temperate regions of central western Europe. Unlike in Mediterranean catchments, this flooding behaviour is unusual. In the past, and especially in the 1990s, floods were characterized by predictable, slowly rising water levels during winter and driven by westerly atmospheric fluxes. Here, we explore potential links and causes between the recent occurrence of flash floods in central western Europe to extreme 15 precipitation and specific atmospheric conditions. We hypothesise that a change in atmospheric conditions led to more frequent extreme precipitation events that subsequently triggered flash flood events in central western Europe. To test this hypothesis, we compiled data on flash floods in central western Europe and selected precipitation events above 40 mm h$^{-1}$ from radar data (RADOLAN, DWD). Moreover, we identified proxy parameters representative for extreme precipitation favouring atmospheric conditions from the ERA5 reanalysis dataset. High specific humidity in the lower troposphere ($q \geq 0.004$ kg kg$^{-1}$ 20 ), sufficient latent instability (CAPE $\geq 327$ J kg$^{-1}$) and weak wind speeds (WS$_{10m-500hPa} \leq 6$ m s$^{-1}$) proved to be characteristic for intense rainfall that can potentially trigger flash floods. We relied on linear models for analysing 40 years-worth (1981-2020) of atmospheric parameters, as well as related precipitation events. We found significant increases in atmospheric moisture contents and increases in atmospheric instability. Parameters representing the motion and organisation of convective systems remained largely unchanged in the considered period (1981-2020). However, the number of precipitation events, their 25 maximum 5-minute intensities as well as their hourly sums were characterized by large inter-annual variations and no trends could be identified between 2002-2020. Our study shows that there is no single mechanistic path leading from atmospheric conditions to extreme precipitation and subsequently to flash floods. The interactions between the processes involved are so intricate that more analyses are required, considering other potentially relevant factors such as intra-annual precipitation patterns or catchment specific parameters.

# 1 Introduction

Flash floods mostly originate from deep moist convection and rank among the most destructive hazards, leading to economic losses, damage to infrastructure, and high mortality rates (Gaume et al., 2009; Hall, 1981; Llasat et al., 2014; WMO, 2017). They are often accompanied by massive erosion and other geomorphologic processes, such as landslides (Bucała-Hrabia et al., 2020; Vogel et al., 2017). While flash floods remain rather exceptional, their occurrence has more than doubled in Europe since the beginning of the 21$^{st}$ century in comparison to the late 1980s (Marchi et al., 2010; Owen et al., 2018). Flash floods in central western Europe typically affect relatively small areas (a few to 100 km$^2$) and generally last less than seven hours (Marchi et al., 2010). Caused by conditionally unstable atmospheric conditions mainly between May and July, they do not substantially affect the annual water balance. High pre-event soil moisture – caused by rainy weather in the preceding days – may lead to a rapid saturation of soils and a swift onset of extreme runoff response (Marchi et al., 2010). Examples of flash floods in recent years relate to Luxembourg around June 2018 (Pfister et al., 2020) and July 2016 (Pfister et al., 2018), to Braunsbach (Germany) in May 2016 (Bronstert et al., 2017, 2018) or the Starzel river flood in June 2008 (Ruiz-Villanueva et al., 2012).While large scale winter inundations were the most common flood type in western Europe until the 1990s (Pfister et al., 2004), flash flood events have increasingly occurred over the last 15 years (Göppert, 2018; Marchi et al., 2010). This raises the question about the origin of this change in flooding type (Bertola et al., 2020, 2021). In this study, we conjecture that changes in the average atmospheric conditions may lead more often to flash flood prone meteorological conditions.

The definitions of flash floods are manifold and sometimes even equivocal in literature. In this study we focus on pluvial floods triggered by intense (convective) rainfall during summer – typically lasting for a few hours. The response times to peak discharge lie within a similar range. The flood characteristics refer to a comprehensive set of extreme and small-scale floods with rapidly rising and falling limbs of the hydrograph and a high impact in terms of damage to infrastructure and/or casualties in the worst case. The largest floods in our data base involved catchments (Ernz Blanche, Starzel river) with a size just over 100 km$^2$, the smallest events have affected hillslopes of a few 100 meters, where major surface runoff had been reported. We prefer to keep the definition simple and not precisely quantify or limit it to specific processes, as little is understood about the underlying processes. The National Weather Service of the US defines a flash flood similarly broad as 'a rapid and extreme flow of high water into a normally dry area, or rapid rise in a stream or creek above a predetermined flood level, beginning within six hours of the causative event' (NWS, 2021).

Precipitation events potentially causing flash floods are characterized by high rainfall amounts over a sufficient period. This condition is met by high rainfall intensities, which typically last between 30 minutes and a few hours (Doswell et al., 1996; Markowski and Richardson, 2010). This is mostly the case during rainfall events of convective origin. In particular, slow-moving or quasi-stationary multicellular storms can combine both, high rainfall intensities and a sufficiently long duration. Combined effects of several physical processes can cause the most severe rainfall, eventually initiating flash floods. One of these effects consists of storm training, where the storm cells move consecutively in line-parallel direction over the same area, which may then cause high precipitation totals. Another comparable effect leading to abundant precipitation, or a prolonged

event duration is the so-called effect of back building with the forward movement cancelled out by continuous backward development of new cells, leading in the end to a slow ground-relative movement of the whole precipitation area. During the flash flood events in Luxembourg in 2016 and 2018, upscale growth also had a distinctive impact on the precipitation processes (Mathias, 2019, 2021). As a result of this merging of two or more individual convective cells to form a multicell storm, the initial raindrop sizes and dynamics of merging cells are often varied, which then in turn can cause downdrafts producing extremely high precipitation intensities (Doswell et al., 1996; Markowski and Richardson, 2010).

Atmospheric conditions associated with excessive convective rainfall have three major characteristics: (1) sufficient latent instability, (2) high moisture content and (3) a slow storm motion (Van Delden, 2001; Doswell et al., 1996; Markowski and Richardson, 2010; Taszarek et al., 2021a). For deep moist convection to occur, first, the tropospheric lapse rates need to be sufficiently steep and a lifting mechanism is required (Van Delden, 2001). Second, the moisture content in the boundary layer needs to be abundant in order to supply water vapour for condensation during the lifting process. High to moderate values of relative humidity in the lower to mid troposphere can further nurture convective cells through limiting water vapour losses due to evaporation and entraining dry air around convective cell boundaries (Doswell et al., 1996; Markowski and Richardson, 2010; Púčik et al., 2015). The same effect – limiting the diminishment of specific humidity by entrainment – is realized by a wide updraft. Additionally, high freezing levels and low cloud base heights enhance the warm cloud depth and thus allow the warm rain process of collision and coalescence to be more dominant. This leads to a higher precipitation efficiency and is associated with higher rainfall rates (Doswell et al., 1996; Markowski and Richardson, 2010; Schroeder et al., 2016). In continental Europe, high values of total column water vapour are often related to the advection of warm Mediterranean air masses (Van Delden, 2001) or air masses from the subtropical region of the North Atlantic (Mathias, 2021; Mohr et al., 2020). Lastly, to ensure a sufficient duration of the rainfall event, a large rainfall system or slow storm motion is needed (Van Delden, 2001). This generally occurs in case of very weak pressure or geopotential gradients when the mean wind speed and the bulk shear between the surface and the lower to mid troposphere are weak. This process is often enhanced by orography, that influences the near-surface wind field channelling convergence zones (Whiteman, 2000). Moreover, a decoupled flow (rapid vertical shift of prevailing wind directions by at least 90 degrees) between the lower and mid troposphere can significantly reduce storm motion in some cases, as analysed by Mathias (2019).

Proxy parameters from climate reanalysis data are regularly used to identify the atmospheric conditions described above during convective events (Brooks, 2009; Groenemeijer and van Delden, 2007; Púčik et al., 2015; Taszarek et al., 2017; Westermayer et al., 2017). The main parameters used in these studies are the bulk wind shear to estimate the thunderstorm cell organisation and precipitation efficiency, and convective available potential energy (CAPE), to identify atmospheric instability. Púčik et al. (2015) and Westermayer et al. (2017) found heavy precipitation to occur across a wide range of deep-layer wind shear (DLS; bulk shear between the surface and 6 km height). CAPE, as a proxy for latent instability, needs to be reasonably high for thunderstorms to develop (Púčik et al., 2015; Westermayer et al., 2017). When focusing on heavy precipitation events within the range of thunderstorms, high specific or relative humidity are parameters to identify moisture content at different atmospheric levels (Púčik et al., 2015; Westermayer et al., 2017). So far, studies only included the wind speed in the form of

wind shear as a proxy parameter for the potential organisation of convective systems, which is important for hail, severe gusts, and tornadoes. However, the development of flash floods relies on longer-lasting, extreme precipitation. Therefore, the storm motion must be slow, which is dependent on a weak flow in the lower to mid troposphere. Hence, we consider the wind speed as a relevant parameter when assessing the flash flood hazard via a slow storm motion.

The identified atmospheric parameters can be analysed over a longer period for trends or oscillations. Therein, especially trends in atmospheric instability are debated. While several studies find increasing CAPE in reanalysis data, recent studies by Rasmussen et al., 2020, Chen et al. (2020) and Taszarek et al. (2021) point out that CAPE is opposed by increasing convective inhibition (CIN). However, higher CIN levels may lead to higher CAPE values since it prevents premature initiation of convection potentially inhibiting the development of stronger CAPE, and thus possibly increasing the potential of more intense storms. In contrast, decreasing relative humidity levels at low levels of the atmosphere, connected to rising temperatures, could potentially reduce the number of thunderstorms (Taszarek et al., 2021a). Absolute humidity is however expected to increase in warmer conditions and can potentially release higher precipitation totals (Lenderink and Van Meijgaard, 2008; Martinkova and Kysely, 2020; Mishra et al., 2012). Changes in wind shear were found to be minor (Rädler et al., 2018). Rädler et al. (2018) concluded, that the frequency of thunderstorms did not increase significantly over the past 40 years in central western Europe, but that they are more likely to produce severe weather.

So far, most studies have focussed on thunderstorm conditions in general or convective hazards related to lightning, hail, tornadoes, or wind gusts. Here, we focus on the thunderstorm events that cause extreme precipitation and especially flash flood events. Forecasting potential heavy precipitation based on atmospheric conditions remains a major challenge, as different atmospheric constellations (e.g. back-building multicells, chaotic cell clustering, atmospheric rivers) can cause heavy precipitation events, while large hail, for example, is mostly associated with supercells, and therefore less challenging to identify (Púčik et al., 2015).

In view of these recent findings, we hypothesise that a change in atmospheric conditions led to more frequent extreme precipitation events that subsequently triggered flash flood events in central western Europe. Prior to hypothesis testing, we have compiled a comprehensive set of 20-40 years-worth hydro-climatological observation series – including extreme precipitation events, related atmospheric conditions, and documented flash flood occurrences. We then leveraged this dataset for investigating a potential increase in extreme precipitation events in central western Europe. Secondly, we relied on proxy parameters, such as CAPE, specific humidity, and wind speed, for identifying the atmospheric conditions that had prevailed during extreme precipitation and related flash flood events. Third, we applied a trend analysis to the identified set of atmospheric parameters using the ERA5 reanalysis data (Hersbach et al., 2020) for the past (1981-2020). The overarching goal of our study is to contribute to a better understanding of climate change effects, as expressed through modifications in the frequency and severity of extreme precipitation events in a temperate climate – more specifically in an area where flash floods used to be an extremely rare phenomenon until recently.

## 2    Data and Methods

### 2.1    Study area and period

Our study area comprises central western Europe (50.5° N, 10° E, 47.5° S, 5° W) including Luxembourg, south-western Germany, and north-eastern France (Figure 1 a-c). The study period spans the summer months from May to August, that exhibit the most favourable conditions for thunderstorms and the onset of flash floods (Van Delden, 2001; Rauber et al., 2008), between 1981 and 2020.

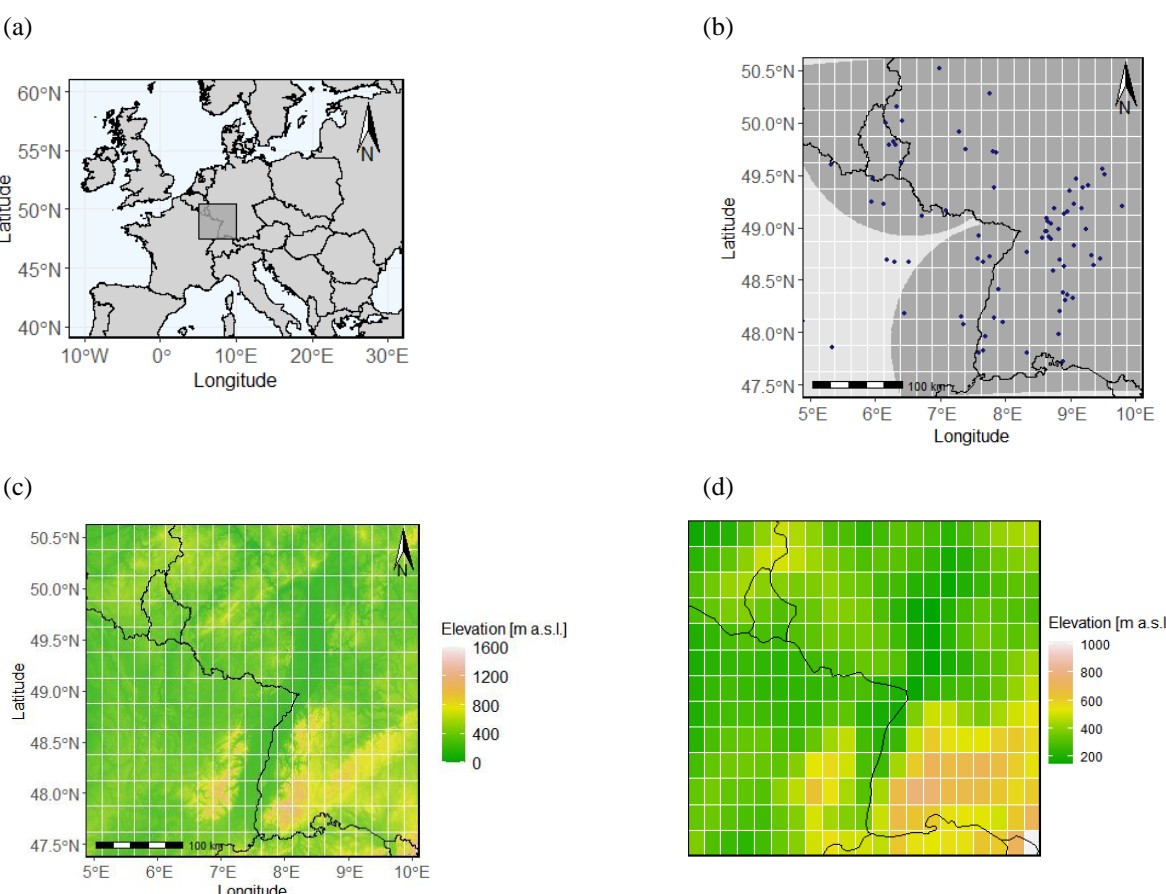

**Figure 1: (a) Location of the study area (dark grey square) within Europe. (b) Map of the study area including data points of occurred flash floods and the range of the DWD-RADOLAN precipitation radar data in dark grey. The white grids show the grid width of the ERA5 reanalysis dataset. Panel (c) shows a digital elevation model of the area of a 1 km x 1 km resolution. (d) Model-topography within the study area based on the ERA5 surface geopotential.**

### 2.2    Database

We downloaded the ERA5 *atmospheric reanalysis data* from the Copernicus' Climate Data Store (CDS) on single levels (Hersbach et al., 2018b) and on different pressure levels (Hersbach et al., 2018a). In addition, we downloaded land data from ERA5 (Muñoz Sabater, 2019) to analyse the pre-event wetness state of soils in catchments. Within the summer months from

May to August for the period 1981 to 2020, selected parameters (cf. Sect. 2.3) were retrieved at a 1-hourly timestep. The horizontal grid spacing of the atmospheric data is 0.25° x 0.25° and of the land data is 0.1° x 0.1°.

The *extreme precipitation event database* was created based on the Radar-based Precipitation Climatology (RADKLIM) dataset from the German Weather service (Version 2017.002 – Winterrath et al., 2018). This is a processed version of the operational RADOLAN radar dataset from the German Weather Service (Weigl et al., 2004; Weigl and Winterrath, 2009; Winterrath et al., 2017). Data are available from 2001-2020 and were considered from May to August. The dataset has a 1 km x 1 km grid size and a temporal resolution of 5 minutes. Unfortunately, the south-western part of the study area is not covered

by the RADOLAN data (Figure 1 b). Although the original RADOLAN product is already quality checked and corrected and consequently reaching high quality, we applied some additional quality control and correction when needed. This included – next to a thorough visual check of the data – the detection and correction of possible anomalous propagation (anaprop) echoes, further ground clutter detection and removal as well as an extended rain gauge adjustment with supplementary local rain gauges. The last operation was done to achieve a further densification of the measuring network (in comparison to the original

product), which is especially important when dealing with flash floods, which often exhibit large spatial precipitation sum gradients. To ensure a comparable standard, we used the same methodology for the rain gauge adjustment as used for the generation of the original RADOLAN/RADKLIM dataset, that is the best combination of the multiplicative and the additive adjustment (Bartels et al., 2004; Wilson and Brandes, 1979; Wood et al., 2000). The adjustment interval was one hour. The extra stations used, were – in Luxembourg – mainly the stations of the ASTA network (Administration des Services Techniques

de l'Agriculture) (ranging from 7 to 40 extra stations), and – in Germany – the stations of the agricultural-meteorological network of the state of Rhineland-Palatinate (ranging from 10 to 50 extra stations). The additional rain gauge data was quality controlled based on Sevruk (1985) and Michaelides (2008). We extracted the precipitation events (P events) for the database from the radar database by identifying 1 km x 1 km grid cells with precipitation amounts $\geq$ 40 mm h$^{-1}$. Connected grid cells with maximum one cell (1 km distance) in between two or more cells exceeding the threshold and a maximum of half an hour

time gap were clustered to account for one P event (Figure 2). The threshold of 40 mm h$^{-1}$ was used according to the definition of extreme precipitation events by the German Weather Service (DWD, 2021). This approach led to a total of 3835 P events between 2001-2020 (Table 1). For every P event, we extracted the maximum hourly precipitation intensity as well as the maximum 5-minute precipitation intensity at one location within the P event. Moreover, the temporal (time of the first threshold exceedance in one of the grid cells of the P event to the time of the last exceedance) and spatial (area of the number of grid

cells that are part of a P event) distribution of the events were identified. Atmospheric conditions during P events were identified at the beginning of a P event, as atmospheric conditions should be the most characteristic at the onset of a P event. To receive a spatially representative value the mean was calculated of each atmospheric ERA5 grid cell of the P event itself as well as a buffer zone around according to the schematic representation in Figure 2. For a small standard P event that lies within one ERA5 grid cell, atmospheric data was averaged over that particular ERA5 grid cell and the eight surrounding ones.

Precipitation events at the boundary of the study area do not include the full buffer zone and larger P events covering multiple

grid cells include a buffer zone around the ERA5 grid cells of the actual P event. A more detailed description of this procedure and its special cases are documented in the Supplement S3.

The *flash flood database* was compiled via a search through case studies in scientific literature (Brauer et al., 2011; Bronstert et al., 2018; Van Campenhout et al., 2015; Eden et al., 2018; Göppert, 2018; Ruiz-Villanueva et al., 2012), water agency reports (Johst et al., 2018; Pfister et al., 2018, 2020), reinsurance data (Caisse Centrale de Réassurance (CCR), 2021), personal communication (engineering consultants Wald + Corbe) and news archives (Franceinfo, 2021; Luxemburger Wort, 2021). We included floods in streams, fields or on streets that are spatially (max. 30 km) and temporally (same day) linked to an extreme P event exceeding the threshold of 40 mm $h^{-1}$. If a flood was triggered by a rainfall event not identified as extreme in the radar data, the flood was not considered. Despite a careful and comprehensive query, the database is likely non-exhaustive. Yet, we think that this approach of site inspections is the most inclusive. Sufficient discharge time series are mainly available for larger rivers and bigger stream gauges, than the ones in which flash floods occur. Moreover, data availability in the past is often limited to a daily resolution, which can easily miss capturing peak flows during flash floods. Relying on high flow water levels in the past makes it also difficult to distinguish flash floods from slowly developing floods, which occurred regularly in the past, especially in the mountainous parts of the study area. A particular example of limits of a discharge time-series based approach are the flash floods in Luxemburg (Pfister et al., 2018, 2020), which were detected by stream gauges only to a limited extent within an overall time series that is too short for any long-term analyses, A list of the 40 events that were eventually included in this study spanning the period from 2002 to 2020 can be found in the Supplement S1 of this manuscript. To extract atmospheric parameters during flash flood events, we identified the triggering P event within a 30 km range and proceeded according to the approach for P events as shown in Figure 2. With this approach, we found 37 from the total of 3835 P events, that are associated with flash floods (Table 1). These are less than the number of flash floods themselves, as in 2008, two flash floods were triggered by the same P event (Rangendingen, Jungingen, Ruiz-Villanueva et al., 2012) and in 2018 three floods were triggered by the same P event (Rhineland-Palatine, Johst et al., 2018).

**Table 1: Total number of P events and number of P events among which were associated with flash floods.**

|  | P events | P events associated with FF | FF events |
|---|---|---|---|
| No. of events | 3835 | 37 | 40 |

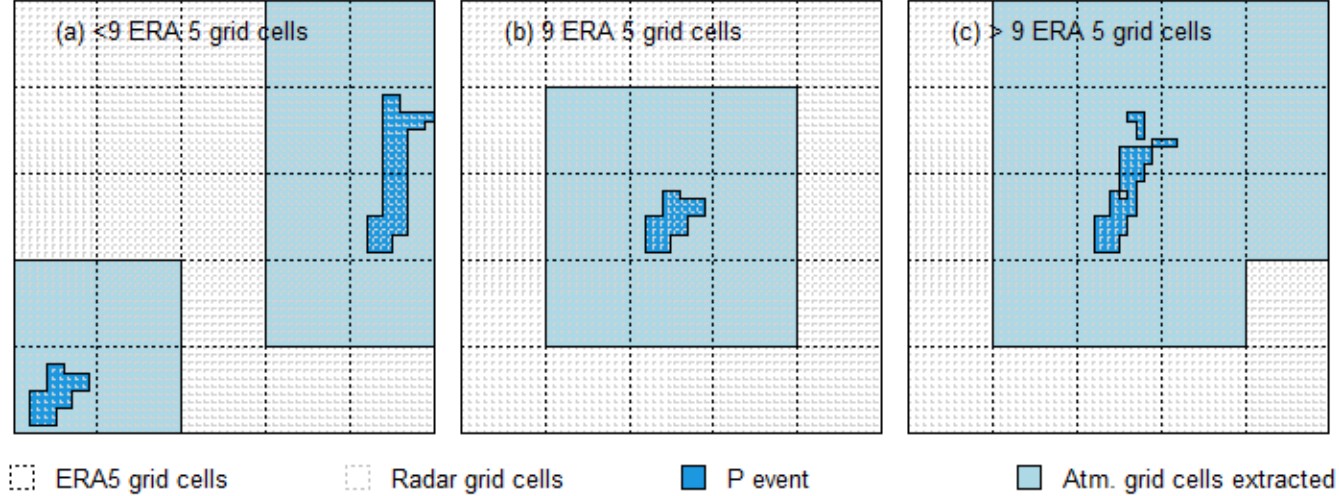

**Figure 2: Schematic representation of the ERA5 grid cells (0.25° x 0.25°, ~ 25 km x 25 km) that were averaged to calculate representative atmospheric conditions during P events (grid width 1 km x 1 km). Marked with dashed lines is the grid of the ERA5 cells (black) and the radar cells (grey). The radar grid cells marked in blue are the ones exceeding the precipitation threshold during a P event. The surrounding ERA5 grid cells marked in light blue are the ones that were used to average the atmospheric conditions Panel (b) shows the standard case of a buffer zone of a one ERA5 grid cell P event, while panel (a) shows some possible exceptions at the boundary of the study area and panel (c) shows the procedure for larger P events covering multiple ERA5 grid cells.**

### 2.3    Identification of atmospheric parameters favouring extreme precipitation and flash floods

Referring to work done by Van Delden 2001; Westermayer et al. 2017; Taszarek, Brooks, and Czernecki 2017; Púčik et al. 2015, we selected relevant atmospheric parameters to represent (1) instability, (2) the moisture content, and (3) the storm motion. Additionally, we extracted (4) soil moisture content from the ERA5 dataset to get an indication for the wetness state of the catchment before the onset of an extreme precipitation event (Table 2).

(1) As proxy parameters for atmospheric instability, we used the convective available potential energy (CAPE) [J kg$^{-1}$], which is provided within the ERA5 single level datasets. In addition, we also considered convective inhibition (CIN) [J kg$^{-1}$]. Given its recognised potential as flash flood proxy, we used the K-Index [°C] that is provided within the ERA5 dataset. The K-Index (George, 1960) is defined via Eq. (1) where $T$ is the air temperature at differing pressure levels and $Td$ the dew point temperature in °C.

$$K\text{-}Index = (T_{850\ hPa} - T_{500\ hPa}) + Td_{850\ hPa} - (T_{700\ hPa} - Td_{700\ hPa}) \tag{1}$$

The K-Index is a stability index, based on the vertical extent of low-level moisture and the vertical temperature lapse rate of the lower and mid-troposphere. While the operational use of stability indices alone is limited (Doswell and Schultz, 2006), indices can provide additional value when assessing severe weather potential. The K-Index was originally developed to assess potential air mass thunderstorms, or thunderstorms without a dynamic triggering mechanism (George, 1960). Most importantly, it shows some special skill in forecasting the potential of thunderstorms related to heavy precipitation (Funk, 1991; Junker et al., 1999). Regarding the potential for heavy

precipitation, it can be generally stated that the higher the K-Index value, the greater the potential for heavy rain. Originally, K-Index values above 20°C indicate thunderstorms, while there is no thunderstorm potential for values below 20. K-Index values are further subcategorized into isolated thunderstorms (20°C – 25°C), widely scattered thunderstorms (25°C – 30°C), scattered thunderstorms (31°C – 35°C) and numerous thunderstorms (> 35°C). Note that the highest category with K-Index values above 35°C is however extremely rare in central western Europe (< 0.5% within the study area and period, as calculated based on the used ERA5 data).

(2) To reach a sufficiently high rainfall rate causing heavy precipitation and consequent flash floods, the atmosphere's moisture content is pivotal. We opted for the total column water vapour (TCWV) [kg m$^{-2}$] as well as specific humidity (q) [kg kg$^{-1}$] and relative humidity (RH) [%] at the pressure level of 700 hPa as atmospheric moisture content proxies. The pressure level at 700 hPa was chosen, because it is approximately the middle of the lower, weather relevant part of the atmosphere between the surface and 500 hPa.

(3) To assess the storm motion, we computed the wind speed (WS) from the square root of the squared northward direction wind vector u [m s$^{-1}$] and the squared eastward direction wind vector v [m s$^{-1}$] at the pressure level of 700 hPa. In addition, the mean of the wind speed between 10 m above the ground level and the pressure level of 500 hPa was calculated. Low-level wind shear (LLS) [m s$^{-1}$] was likewise computed based on the square root of the differences of the vectors u and v near the ground and at about 1.5 km height levels (850 hPa). Accordingly, we calculated the deep-layer wind shear (DLS) [m s$^{-1}$] as the difference of the wind vectors near the ground and in about 6 km height levels (500 hPa). The wind shear allows an assessment of the organisational mode of deep moist convection.

(4) We considered soil moisture parameters for assessing the pre-event wetness state of a catchment. Therefore, we extracted soil moisture (Swvl) [m$^3$ m$^{-3}$] at depths of 0-7 cm, 7-28 cm, and 28-100 cm from ERA5, 24 hours before the onset of identified P events and the onset of flash flood triggering P events respectively.

**Table 2:** Selected proxy parameters for the assessment of convection relevant atmospheric conditions from the ERA5 dataset.

| Proxy for | Parameter | Abbr. | Unit | Level | Source |
|---|---|---|---|---|---|
| Instability | Convective available potential energy | CAPE | J kg$^{-1}$ | single | Hersbach et al., 2018b |
| | Convective inhibition | CIN | J kg$^{-1}$ | single | Hersbach et al., 2018b |
| | K-Index | Kx | °C | single | Hersbach et al., 2018b |
| Moisture | Total column water vapour | TCWV | kg m$^{-2}$ | single | Hersbach et al., 2018b |
| | Specific humidity | q | kg kg$^{-1}$ | 700 hPa | Hersbach et al., 2018a |
| | Relative humidity | RH | % | 700 hPa | Hersbach et al., 2018a |
| | U-component of wind | u | m s$^{-1}$ | 10 m, | Hersbach et al., 2018a |

| Storm motion and organisation | V-component of wind | v | m s⁻¹ | 500 hPa & 700 hPa | Hersbach et al., 2018a |
|---|---|---|---|---|---|
| Catchment wetness state | Volumetric soil water layer 1 | Swvl1 | $m^3\ m^{-3}$ | 0-7 cm | Muñoz Sabater, 2019 |
| | Volumetric soil water layer 2 | Swvl2 | $m^3\ m^{-3}$ | 7-28 cm | Muñoz Sabater, 2019 |
| | Volumetric soil water layer 3 | Swvl3 | $m^3\ m^{-3}$ | 28-100 cm | Muñoz Sabater, 2019 |

To identify extreme precipitation and flash flood relevant proxy parameters, we extracted their respective values from the
ERA5 atmospheric dataset at the time step and grid cell of initially identified events. Next, we created thresholds for every proxy parameter, that make the occurrence of precipitation events possible. Therefore, we chose the 75th or the 25th percentile as the upper or the lower boundaries including either the lower or the upper three quartiles of all values of extreme events. These percentiles were chosen as statistical standard as also used in Schroeder et al. (2016). This analysis leads to the determination of the thresholds in Table 3, Sect. 3.3, to classify atmospheric conditions as extreme precipitation and potentially
flash flood favouring. We used these thresholds, as well as the three parameters identified the most suitable from the groups of moisture, instability, and storm motion and organisation to eventually conduct trend analyses.

### 2.4 Trend analyses

We carried out linear trend analyses to test the different parts of our working hypothesis – linking a potential increase in atmospheric conditions triggering extreme precipitation events to a rise in the occurrence of extreme precipitation events in
central western Europe. We applied the linear models to our precipitation event database, as well as to the occurrence frequency, precipitation amount and intensity of identified extreme precipitation events. Likewise, we applied linear models to the flash flood relevant parameter ranges of the identified set of ERA5 atmospheric parameters, as well as to the simultaneous occurrences of the three most relevant parameters.

# 3 Results

## 3.1 Flash flood occurrences

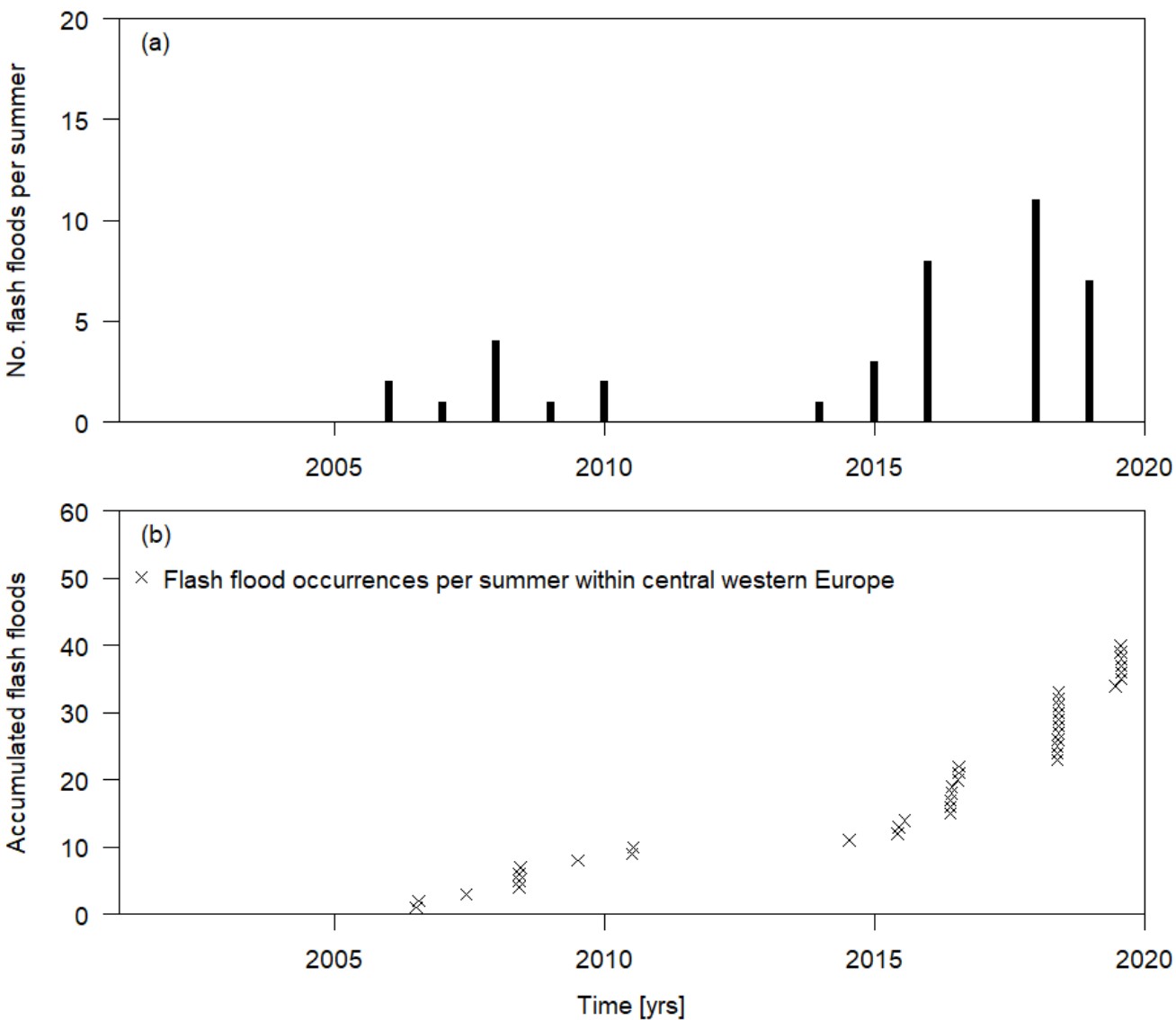

**Figure 3: Occurrence of flash flood events within central western Europe between 2001 and 2020. Panel (a) shows the number of flash flood occurrences per summer, panel (b) maintains the exact occurrence date of the flash flood event.**

Figure 3 shows flash flood occurrences in central western Europe. While barely any events were reported before 2006, two

remarkable summers are 2016 and 2018, when flash floods occurred particularly often in the study area (8 and 11 occurrences respectively). As the temporal inconsistencies in the dataset do not allow drawing conclusions on any robust trends, this flash flood data compilation cannot support the conjectured increase in frequency of flash floods. Note that often several events

occurred within a few days (Figure 3 b) under the same meso-scale atmospheric constellation, in the same area or even in neighbouring catchments, and are, therefore, not completely independent from one another. Two flash floods in 2008

(Rangendingen, Jungingen, Ruiz-Villanueva et al., 2012) and three floods in 2018 (Rhineland-Palatine, Johst et al., 2018) occurred during the same large-scale P event.

### 3.2 Extreme precipitation event characteristics

Within our study area, we extracted extreme P events with precipitation intensities $\geq 40$ mm h$^{-1}$ from the DWD-radar-dataset. Between 2001 and 2020, we observed a slight, but insignificant increase in the number of events per summer (Figure 4a). Note

that interannual variance is very high and that this increase includes two extreme years, 2006 and 2018, when precipitation events $\geq 40$ mm h$^{-1}$ occurred particularly often. Similar to the flash flood occurrences, many of the extreme precipitation events happened on the same days over a wider region. This is particularly the case for 2008 and 2018 – with the multiple rainfall events from 2018 overlapping with a high number of flash floods. For the precipitation amounts we could not identify significant trends in the 2001-2020 period for both the maximum 5-minute precipitation intensities (Figure 4b) and the

maximum hourly intensities per event (Figure 4d). P events that eventually led to flash floods (Figure 4c, e) do not differ in the range of precipitation intensities from P events that did not cause flash floods, yet their median is around 3 mm h$^{-1}$ higher. The event duration of P events that caused flash floods is however slightly longer compared to the other extreme P events (Figure 4f, g). The largest difference between P events causing flash floods and other P events is, however, the temporal and spatial extent. P events that cause flash floods are often longer lasting and larger in comparison to extreme P events that did

not lead to flooding (Figure 4h, i). Neither the temporal nor the spatial extent of the P events shows trends over the study period of 20 years (Figure 4f, h).

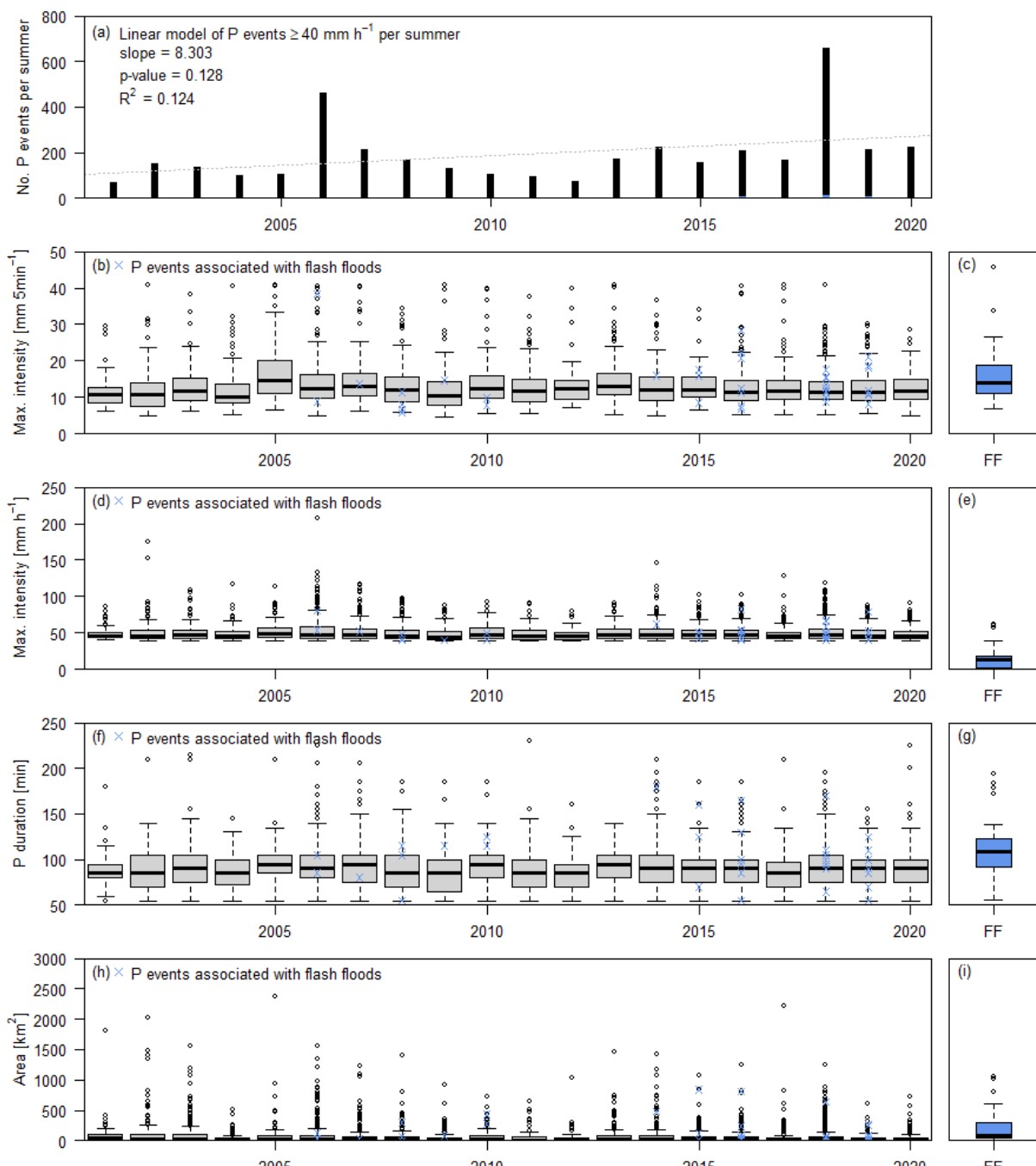

**Figure 4: Occurrence of extreme precipitation events (≥ 40 mm h⁻¹) within central western Europe. The panels in the left column show the precipitation event characteristics per summer between 2001-2020. The blue crosses and the right column (c, e, g, i) show the precipitation characteristics of the events, that are associated with a flash flood. Panel (a) shows the number of precipitation events per summer. The panels (b) and (c) show the P events' maximum precipitation intensity in 5 minutes per event, and per hour (d) and (e). The panels (f) and (g) show the temporal and (h) and (i) the spatial extent of the identified events.**

### 3.3 Identification of atmospheric parameters favouring extreme precipitation and flash flood events

To identify parameter ranges that favour flash floods, we considered all hourly values of the parameters between May and August irrespective of any identified events, as events could only be identified within the last 20 years of the study period. Moreover, we extracted the parameters present during the time of extreme P events and the selection of P events that led to flash flood occurrences (Figure 5). The data emphasise the occurrence of extreme events under conditionally unstable atmospheric conditions. Most extreme precipitation and flash flood events occurred within the upper quartile of CAPE values (Figure 5a). Sufficient values of CAPE are often accompanied by moderate values of CIN. Both extreme precipitation and flash flood events occurred over a wide range of CIN, with a slightly higher median value at the onset of an event compared to the general values (Figure 5b). However, both CAPE and CIN appear to be widely scattered within the spectrum of their possible ranges. The K-Index, in contrast, proves to be a reliable index and more than 80% of all extreme precipitation and flash flood events occur within the thunderstorm relevant categories of the index above 28°C (Figure 5c). Moisture conditions during extreme precipitation and flash flood events were found to be mostly within the upper percentiles of the overall simulated values. Especially the specific humidity (q) and total column water vapour (TCWV) range clearly within the upper quartile of all values during events (Figure 5d, e). Relative humidity (RH) also proves to always be high during extreme events (Figure 5f). All moisture parameters, and especially RH tend to be even higher during flash flood events compared to general extreme precipitation events (Figure 5d-f). The wind related parameters considered to analyse storm motion and organisation are generally low during extreme precipitation and flash flood events. Especially the $WS_{10m-500hPa}$ (Figure 5h) stands out, with most of the values observed during extreme events being in the lower quartile of the full range of occurrences. Tendencies regarding $WS_{700hPa}$, DLS and LLS (Figure 5g, i, j) are less clear but show the same pattern. In addition to atmospheric parameters, soil moisture conditions were evaluated 24 h before identified events. Often, soil moisture within the upper and lower soil layer ($Swvl1_{0-7 cm}$, $Swvl3_{7-100 cm}$) is higher during flash flood events compared to general extreme P events (Figure 5k, m). Especially the higher top level soil moisture might hint to preceding rainfall events that could help explaining some of the quick runoff formation present during flash floods. The mid-level soil layer ($Swvl2_{7-28 cm}$) shows lower soil moisture before flash flood events (Figure 5l).

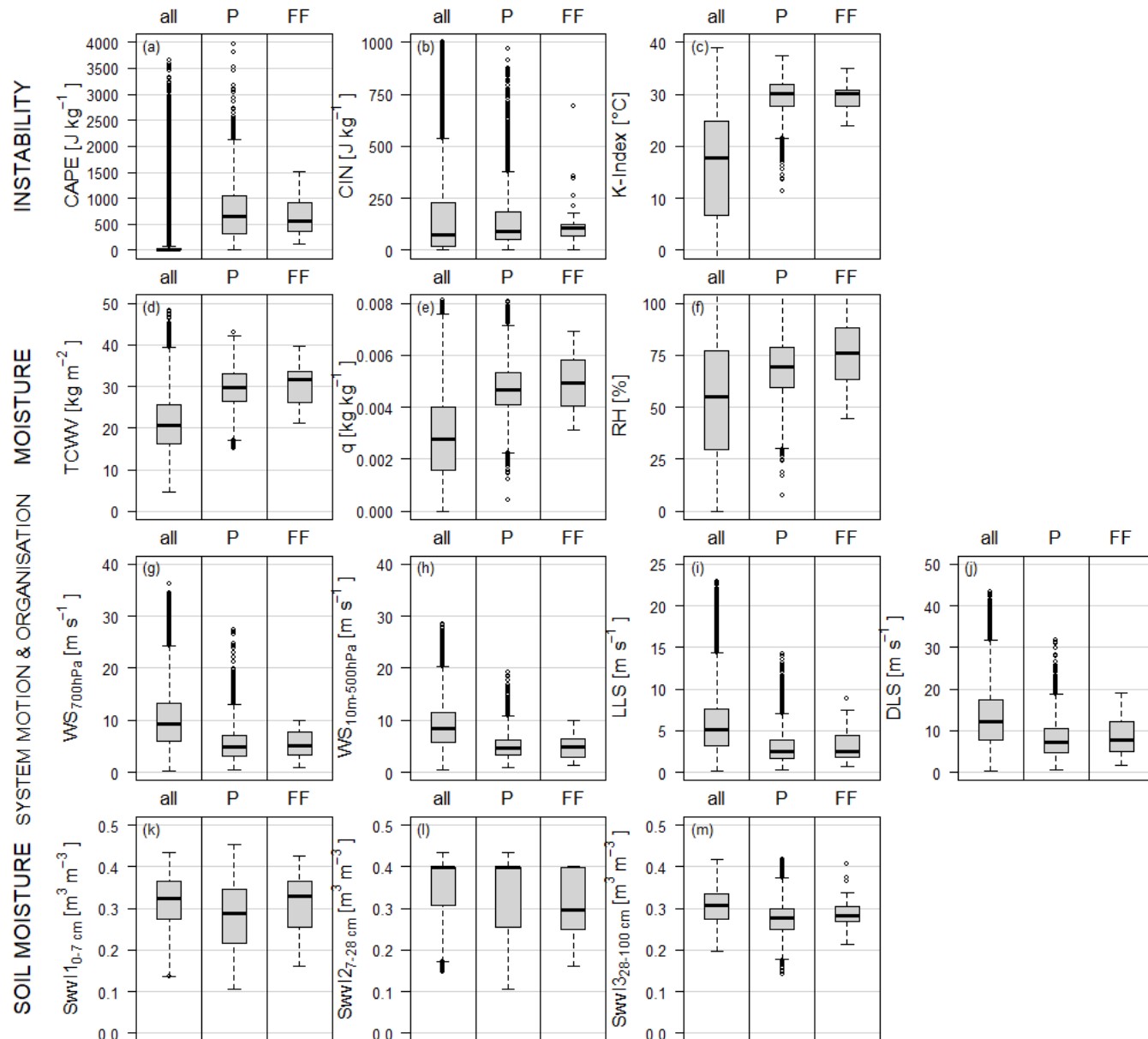

**Figure 5: All hourly values of the proxy parameters (a-j) during the entire period (all), before extreme precipitation events (P) and before flash flood events (FF). Soil moisture (k-m) was extracted 24 hours before the onset of identified P events or 24 hours before the onset of FF triggering P event.**

This analysis leads to the determination of the thresholds in Table 3 to classify atmospheric conditions as extreme precipitation and potentially flash flood favouring. Sufficient CAPE, high q and weak $WS_{10m-500hPa}$ were identified as the most clearly distinguishing parameters per category to characterize extreme precipitation events, including 75% of all extreme precipitation events and excluding around 75% of all generally occurring parameters values.

**Table 3:** Threshold values determined as extreme precipitation and flash flood favouring based on the lower/upper quartile of their range of occurrence during extreme precipitation events, including all P events, whether they are associated with a flood or not.

| Instability | | | Moisture | | | Storm motion & organisation | | | |
|---|---|---|---|---|---|---|---|---|---|
| CAPE | CIN | Kx | TCWV | q | RH | $WS_{700\,hPa}$ | $WS_{10m\text{-}500hPa}$ | LLS | DLS |
| $\geq 326.9$ | $\leq 183.5$ | $\geq 27.8$ | $\geq 26.5$ | $\geq 0.004$ | $\geq 59.4$ | $\leq 7.1$ | $\leq 6.2$ | $\leq 3.8$ | $\leq 10.4$ |
| J kg$^{-1}$ | J kg$^{-1}$ | °C | kg m$^{-2}$ | kg kg$^{-1}$ | % | m s$^{-1}$ | m s$^{-1}$ | m s$^{-1}$ | m s$^{-1}$ |

### 3.4 Changes of atmospheric parameters between 1981-2020

Instability, as shown representatively by CAPE above 326.9 J kg$^{-1}$, has increased between 1981 and 2020. The number of hours with high enough instabilities to support the occurrence of thunderstorms increased by up to five hours per summer (Figure 6a). These findings were particularly significant in the northern part of the study area (Figure 6b). There are moreover significant increasing trends regarding the actual values of CAPE above 326.9 J kg$^{-1}$ in the north-western and mid-southern part of the study area (Figure 6c, d). Another measure for the atmosphere's instability and capability to produce rain-intense thunderstorms is the K-Index, shown in Figure A1. The occurrence of K above 27.8°C is strongly increasing between 1981 and 2020 throughout the study area and is significant in the northern part of it. Moreover, the values of the K-Index above the threshold have increased, which indicates an increased intensity of rain-intense thunderstorm events. This trend is significant over the Belgian part of the study area.

The observed increase in high atmospheric moisture content, represented by specific humidity above 0. 004 kg kg$^{-1}$ (Figure 6e, g), is highly significant over the entire study area (Figure 6f, h). Conditions with high moisture content became up to 8 hours per summer more frequent, especially over south-western Germany. The absolute increase of very high moisture content is however small (Figure 6g).

The storm motion potentially decreases with weak $WS_{10m\text{-}500hPa}$ that tends to occur more often in the study area (Figure 6i). The values below the threshold of 6.2 m s$^{-1}$ appear to become higher in the western part of the study area and lower within the eastern part. These trends are however insignificant over the entire study area (Figure 6j, l) and $WS_{10m\text{-}500hPa}$ is considered to remain largely unchanged.

The complete set of analysed atmospheric parameters is shown in Appendix A.

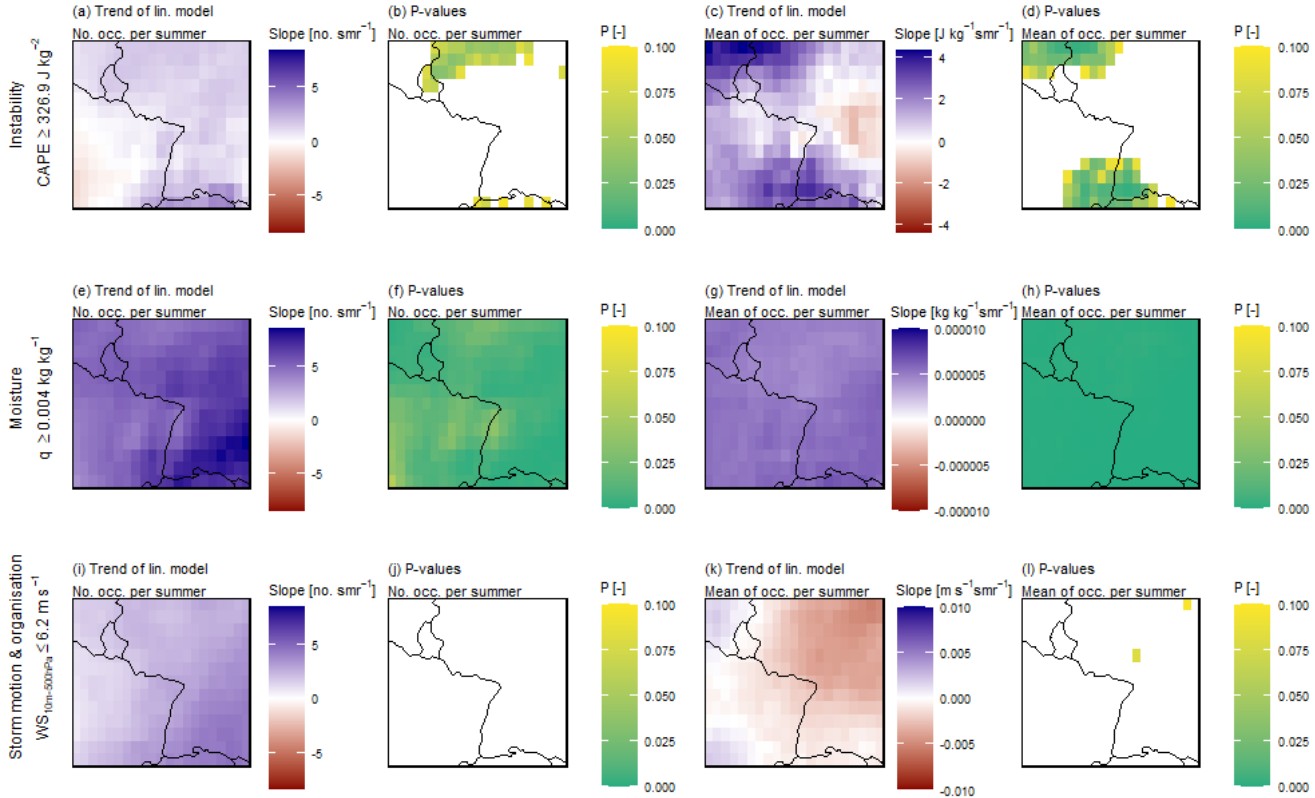

**Figure 6:** Trend analysis of the most suitable variables for instability (CAPE), moisture (specific humidity q), and storm motion and organisation (wind speed, $WS_{10m–500hPa}$). The first column (a, e, i) shows the trends of the numbers of hourly occurrences of values above or below their respective threshold, including their significance-levels in the second column (b, f, j). The third column (c, g, k) shows the trends of the mean values of all hourly occurrences above or below the threshold and the last column (d, h, l) their respective significance-levels. White areas mark insignificance.

### 3.5 Spatial distribution of atmospheric conditions favouring extreme precipitation and flash flood events

The simultaneous occurrence of the three most characteristic identified atmospheric parameters from each component (CAPE, q, $WS_{10m–500hPa}$) within extreme event favouring parameter ranges is correlated with topography (Figure 1c). Favourable atmospheric conditions occur most frequently over the Vosges Mountains in France and in south-western Germany, compared to the rest of the study area. Over eastern Belgium, favourable atmospheric conditions have occurred less than half as often between 1981 and 2020 (Figure 7a). Within this period, the occurrence of favourable atmospheric conditions changed very little. Over south-western Germany, the simultaneous occurrence of these three parameters occurred only 1-2 h per summer more often, while over north-eastern France these conditions occur slightly less often (Figure 7b). There is, however, no significance of trends regarding these combinations (Figure 7c). Splitting the 40-year period in two, 1981-2000 (Figure 7 d-f) and 2001-2020 (Figure 7 g-i), shows a decreasing trend within the first 20 years and a positive trend within the last 20 years. As these seem to be clear tendencies, they more or less level out over the entire time period. In line with the large variation of

the number of occurrences of favourable atmospheric conditions per summer, none of the calculated trends are significant
(Figure 7 f, i).

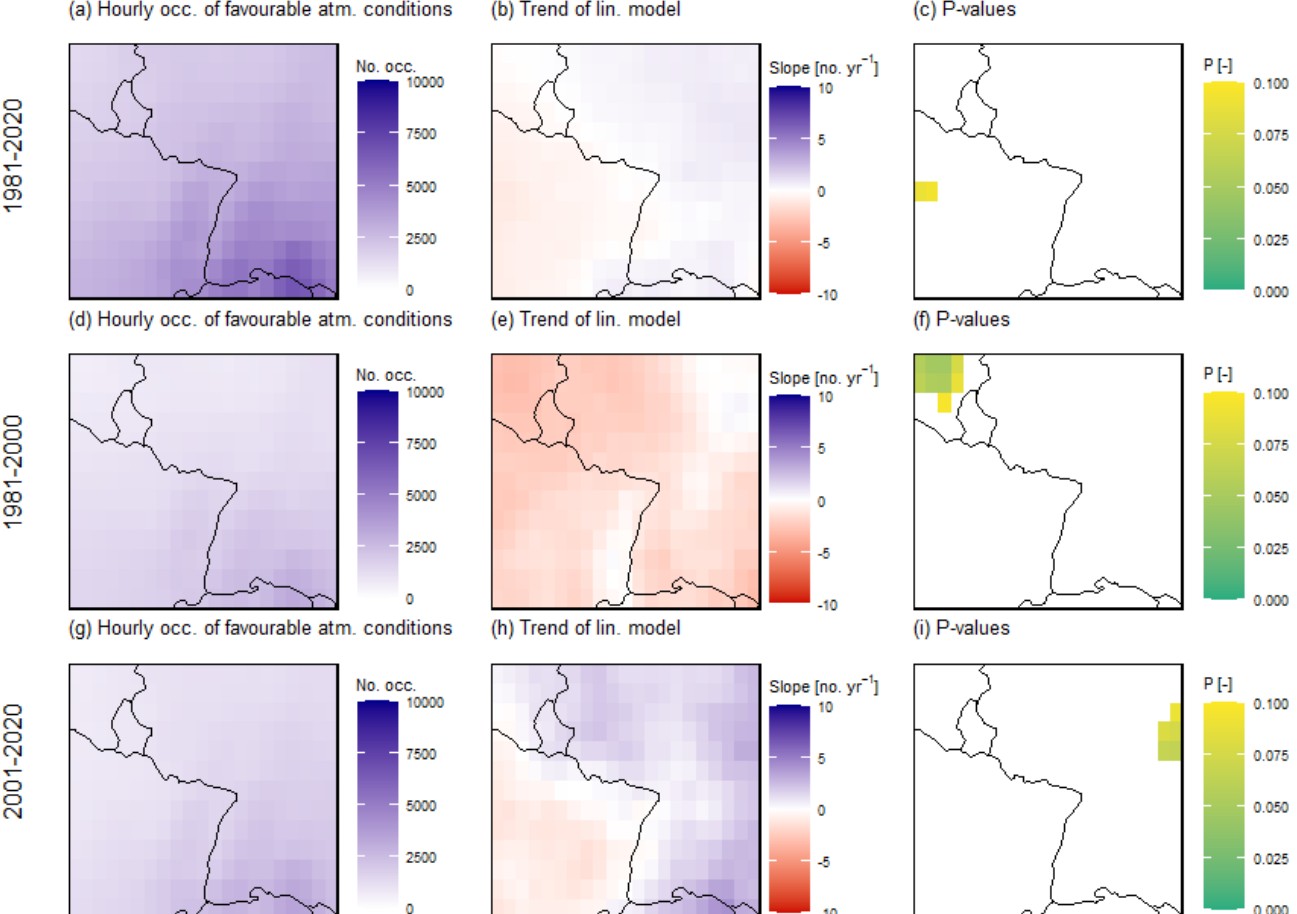

**Figure 7:** Panel (a) shows the overall number of hourly occurrences of atmospheric conditions favouring extreme precipitation and flash flood events during the summer months between 1981 and 2020. Panel (b) illustrates the positive trends of atmospheric conditions favouring extreme precipitation and flash flood events per year, and panel (c) the significance of the linear model. White areas mark insignificance.
The panels (d)-(f) show the same for the time period of 1981-2000, and (g)-(i) for 2001-2020.

## 4    Discussion

We numbered P events as one event when the temporal distance between two events exceeds half an hour, and the spatial distance 2 km. This method does not always account for connected events, such as back-building effects of thunderstorm cells and may lead to an artefact of counting too many P events. In case of several P events per day on which a flash flood was
identified, only the spatially closest or first P event was determined as flash flood triggering which may sometimes underestimate the P characteristics leading to a flash flood. These phenomena are however considered neglectable in central Western Europe, where slowly moving single cell thunderstorms are the main cause for flash floods, as indicated by the low

DLS values identified. This characteristic is in contrast with larger events in the Mediterranean (Gaume et al., 2009) or the US (Gochis et al., 2015).

In the Mediterranean area (Llasat et al., 2016) and lowland catchments of Alpine regions increases in flash floods have been observed (Sikorska-Senoner and Seibert, 2020). In central western Europe, there is moreover an increase in the number of reports and scientific publications on flash floods (e.g. Bronstert et al., 2018; Van Campenhout et al., 2015; Marchi et al., 2010; Ruiz-Villanueva et al., 2012). However, as per their nature, flash floods are rare phenomena. Therefore, we are not able to proof any trends based on the data we have collected. The method of data collection is influenced by the progress of

digitalisation which might make recent years appear more often in search engines. Additionally, we browsed through historical archives but did not find further entries. Any identified trend would moreover be strongly influenced by two years in which especially many events occurred: 2016 and 2018 (and possibly the July 2021 floods, that were not considered in this manuscript but may further strengthen a possible increasing trend). During these event series, atmospheric conditions were characterised by exceptionally long-lasting weather patterns associated with very moist and unstable air masses. These conditions led to the

extraordinarily high number and severity of thunderstorms with substantial flooding in central western Europe (Mohr et al., 2020; Piper et al., 2016).

Based on the DWD's RADOLAN dataset we were not able to detect any linear trends in the number of precipitation events per year, their maximum hourly and 5-minute intensities between 2002-2020. These findings are in line with similar analyses done by the DWD and GDV (2019). As the detection of extreme precipitation events remained challenging due to their

localised occurrence, large-scale data were only available through the deployment of a dense radar station network as of 2002. Note that this observation period remains rather short and does not allow to infer solid conclusions on potential trends. Also, while weather radars provide precipitation data of high spatial resolution, various sources of uncertainty may prevail – related to precipitation type and intensity, topography, distance to the radar source, etc. (Meischner, 2014; Strangeways, 2007; Winterrath et al., 2017). We accounted for some of these potential effects (e.g., rain gauge adjustments, detection, and

correction of possible anomalous propagation echoes). Perhaps, trends in extreme precipitation events could be detected when considering preceding decades as well. Müller and Pfister (2011) analysed longer time series starting in 1980 and indeed found an increase in intense rainstorms during summer months in western Germany (Emscher-Lippe catchment). However, precipitation generally varies considerably on an interannual basis and makes trend analyses challenging. In previous work (Meyer et al., 2020), we analysed 98 daily precipitation station data in the Moselle catchment, which is situated in the west of

the study area, over a 65-year period and could not find trends in the daily precipitation maxima as well as the number of days with precipitation amounts above 50 mm d$^{-1}$. While the daily precipitation sum should be a reliable indicator for extreme precipitation amounts, the coarse station network probably missed high rainfall amounts that fell in between stations. As both the long-term coarsely resolved dataset and the highly resolved short-term dataset did not show clear trends, we could not confirm the hypothesis of an increase in extreme precipitation events within the study area.

We found that atmospheric conditions favouring extreme precipitation and subsequent flash flood events became slightly more frequent and intensities of relevant atmospheric parameters increased. The most significant increases were found for the

moisture parameters which are in line with the assumption of the Clausius-Clapeyron relationships (Lenderink and Van Meijgaard, 2008; Martinkova and Kysely, 2020; Mishra et al., 2012). Both, TCWV and q, increased significantly over central western Europe indicating potentially higher precipitation amounts. Yet, rising air temperatures inhibit an increase in higher

RH (Rädler et al., 2018). The increase of q also influences instability parameters, such as CAPE and the K-Index, to increase at a significant level in some areas. This matches well with the findings by Taszarek et al. (2021b) who documented an increase of CAPE over central Europe. Trends of CIN are however ambiguous within the same period. While in some areas favouring conditions do occur more often, there are indications that CIN increases as well. This increase in CIN might offset some of the instability increases through CAPE (Taszarek et al., 2021a). In this study, we did not analyse the simultaneous occurrences of

CAPE and CIN in detail, but Chen et al. (2020) found highly complex interactions, suggesting that future moist convection and rainstorms may become less frequent but more intense. Regarding low wind speeds and weak DLS, we found sightly increasing but barely significant trends. Increasing trends in low LLS are significant in the south-eastern part of the study area. Overall, the proxy parameters used for the assessment of organisation and motion of storm systems stayed largely unchanged with tendencies favouring the occurrence of extreme precipitation. Studies looking at substantial DLS for other convective

hazards such as hail or tornadoes did not identify significant trends in the past over Europe either (Púčik et al., 2017; Rädler et al., 2018). Studies investigating future conditions across the US, however, even suggest decreases in DLS (Brooks, 2013; Diffenbaugh et al., 2013). Wind speed and shear are not directly relevant for triggering precipitation but slightly increasing the duration of an event, they are potentially contributing to the development of flash floods. The coarse resolution of the ERA5 atmospheric data might miss smaller-scale wind features related to orography. Even though extreme precipitation and flash

floods tend to occur locally, they happen during meso- to large-scale favouring conditions, which should be well captured by the reanalysis data.

The values of the considered atmospheric parameters cover the expected ranges of occurrence. However, to include 75% of all precipitation and flash flood events, we had to include an even wider parameter range. This holds especially true for the lower and respectively upper thresholds of CAPE and CIN, that appear low and respectively high compared to common values

present during thunderstorms (Púčik et al., 2015; Taszarek et al., 2017). In the ERA5 data, both parameters showed an extremely high variability in space and time. This variability of CAPE also leads to a relatively low number of hours with all parameters within their ranges, as shown in Sect. 3.5 (Figure 7). What is striking, however, are the consistently low values of DLS. While we stated in the beginning, that DLS can be either low or high, this does not seem the case in this region. Extreme rainfall and flash flood events seem to be consistently caused by slow-moving single-cell thunderstorms. In the US, in contrast,

many flood-producing storms are larger and more organized mesoscale convective systems (Ashley and Ashley, 2008; Schumacher and Johnson, 2006)(Ashley and Ashley, 2008; Dougherty and Rasmussen, 2019; Schumacher and Johnson, 2006). The floods considered in US American studies are however relating to rather large and deadly flash floods, while in central Europe flash floods generally do not reach comparable dimensions.

The focus of our work was the attempt to link atmospheric conditions, extreme precipitation, and flash floods: We hypothesised

that the conjected increase in flash floods is a consequence of more intense or more often occurring precipitation events, that

are initiated by thunderstorm favouring atmospheric conditions. However, reality seems a lot more complex. While atmospheric conditions tend to become more unstable, and overall warmer air masses potentially possess a higher amount of water vapour, the expected increase in (convective) precipitation events were not obvious from the 20 years of analysed data. Other factors than those that we have considered in this study may influence the development of flash floods. One could be the duration of favouring atmospheric conditions. Both remarkable flash flood series from 2016 and 2018 occurred during atmospheric blocking situations (Mohr et al., 2020; Piper et al., 2016) that stymied the movement of the atmosphere, ultimately causing weather constellations to last longer and thus creating extreme situations. In recent years they have been increasingly observed, especially in summer (Detring et al., 2021; Kreienkamp et al., 2021; Lupo, 2020). This could hint to a change in the intra-annual distribution of precipitation, while the number of precipitation events, their maximum 5-minuteand hourly intensity stayed – apart from their large intra-annual variations – at a similar level between 2001 and 2020. Sequences with abundant rainfall may eventually rise a catchment's soil moisture and accelerate the development of a flood event. While the low top-level soil moisture before the precipitation events might show the typical pattern of central Europe, where thunderstorms mostly occur after a few warm and dry days, this does not seem to not be the case, when flash flooding is caused. The soil moisture is then already elevated at the top layers of the soil to the 'average' level by previous rainfall and causes a faster runoff response including infiltration excess overland flow. Flash floods in continental regions mostly occur when soil moisture is high at the onset of an event (Marchi et al., 2010; Pfister et al., 2020). Moreover, catchment-specific parameters such as topography, land use, soil properties, geology or other factors may equally impact the development of flash floods (Marchi et al., 2010).

## 5    Conclusion

The goal of our study was to identify and analyse the atmospheric conditions prevailing during extreme precipitation and flash flood events in temperate regions of central western Europe. For this purpose, we compiled a flash flood database based on scientific literature, water agency data and the information of local consultants and analysed it using linear regression models. For the identification of extreme precipitation events potentially triggering flash floods, we relied on a 5-minute radar dataset (RADOLAN, DWD) and analysed all precipitation events exceeding the threshold of 40 mm h$^{-1}$ statistically considering maximum hourly and 5-minute precipitation intensities as well as the temporal and spatial coverage of events. The identified flash flood and precipitation events were then connected to convection relevant atmospheric parameters of the ERA5 reanalysis dataset representing instability, moisture content and system motion and organisation. We leveraged these data for testing our hypothesis that a change in atmospheric conditions led to more frequent extreme precipitation events that subsequently triggered flash flood events in central western Europe. Note, that the conjectured increase in the occurrence of flash floods could not be tested due to inconsistencies in the data base. We tested our hypothesis in two steps:

I)    An increase in the frequency and intensity of extreme precipitation events could not be supported with the available database and analysis due to a large interannual variation in events and a relatively short period of 20 years. Future

analyses could incorporate the intra-annual temporal distribution of extreme precipitation events. Perhaps, formerly evenly distributed rainfall events tend to occur more condensed within a few days.

II)    Via proxy parameters we did find changes in the occurrence of atmospheric conditions favouring extreme precipitation and flash flood events. High absolute moisture content (specific humidity ($q$), total column water vapour (TCWV)) has increased significantly between 1981 and 2020, while relative humidity (RH) decreased slightly. Proxy parameters representing sufficient instability (CAPE, K-Index) increased as well and so did the convective inhibition (CIN), which might oppose some of the instability gains of CAPE (Taszarek et al., 2021a). Parameters determining

weak storm motion and organisation (wind speed ($WS_{10m-500hPa}$), deep layer shear (DLS)) did not show significant changes, but the occurrence of weak low level shear increased slightly. Overall, the most important components favouring flash flood relevant atmospheric conditions, abundant moisture, and sufficient latent instability, have become more frequent and higher values indicate possibly more severe events.

Consequently, only the sub-hypothesis II is supported, while sub-hypothesis I was rejected. Hence, the simple causal chain

between the atmospheric conditions, extreme precipitation and flash floods assumed in the overarching hypothesis does not do justice to the entire complexity of problems. Interconnections seem far more complex than hypothesised. In addition to the hypothesis, we found mostly higher upper (0-7 cm) and lower (28-100 cm) layer soil moisture during flash flood events compared to general extreme precipitation events. These results might point us in other directions, possibly to changes in intra-annual temporal patterns of rainfall and consequently different pre-event soil moisture conditions. Another explanation might

be non-atmospheric, catchment-specific parameters, that were not considered in this study.

To the best of our knowledge, this work is none the less among the first ones focusing on the convective hazard of extreme precipitation that was often neglected, giving priority to hail or tornadoes. As extreme precipitation is extremely variable in space and time and can derive from many different weather constellations, it remains a challenge to pinpoint atmospheric conditions that trigger them. This makes possible assumptions about the future extremely challenging.

 **Appendix A : Spatial trends of atmospheric parameters within central western Europe**

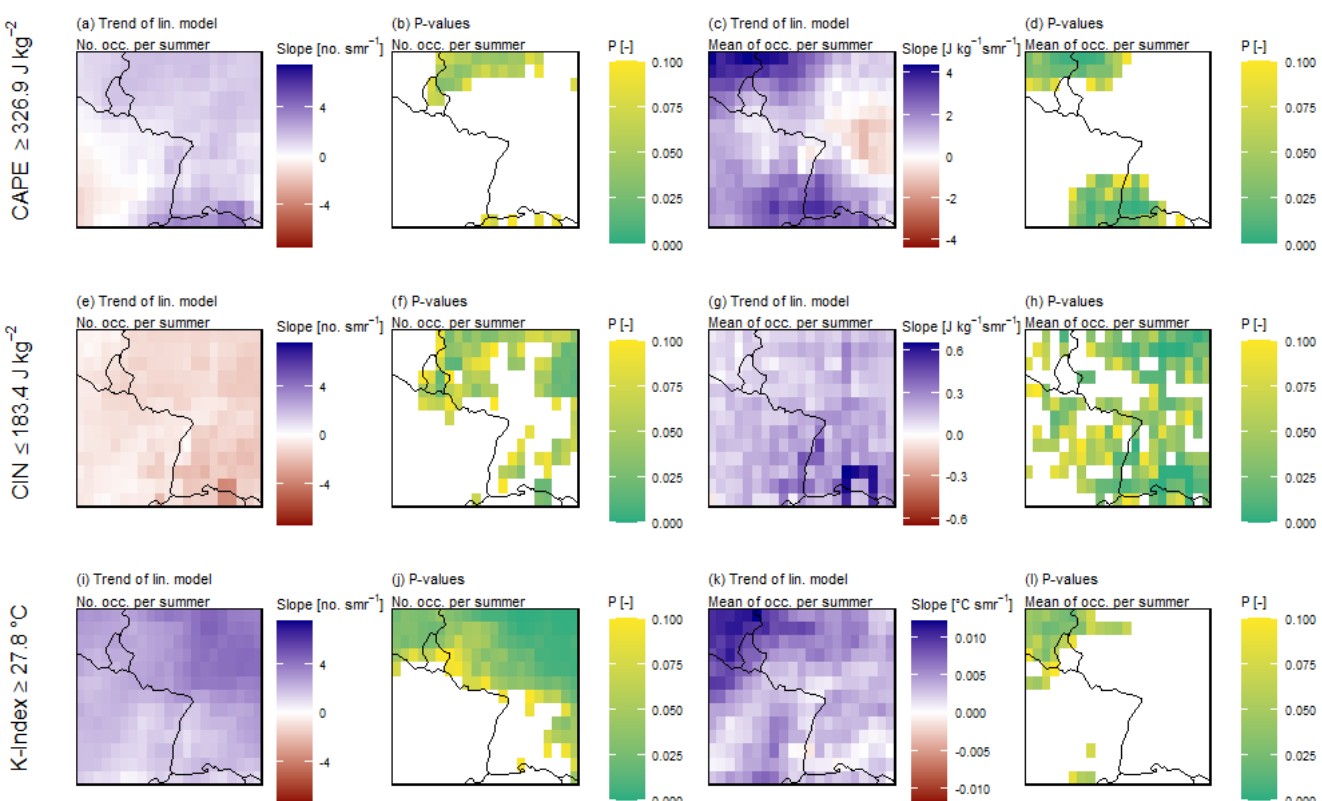

**Figure A1:** Trend analysis of the three variables for instability (CAPE, CIN, K-Index) per summer (smr). The first column (a, e, i) shows the trends of the numbers of hourly occurrences of values above their respective threshold, including their significance-levels in the second column (b, f, j). The third column (c, g, k) shows the trends of the mean values of all hourly occurrences above the threshold and the last column (d, h, l) their respective significance-levels. White areas mark insignificance.

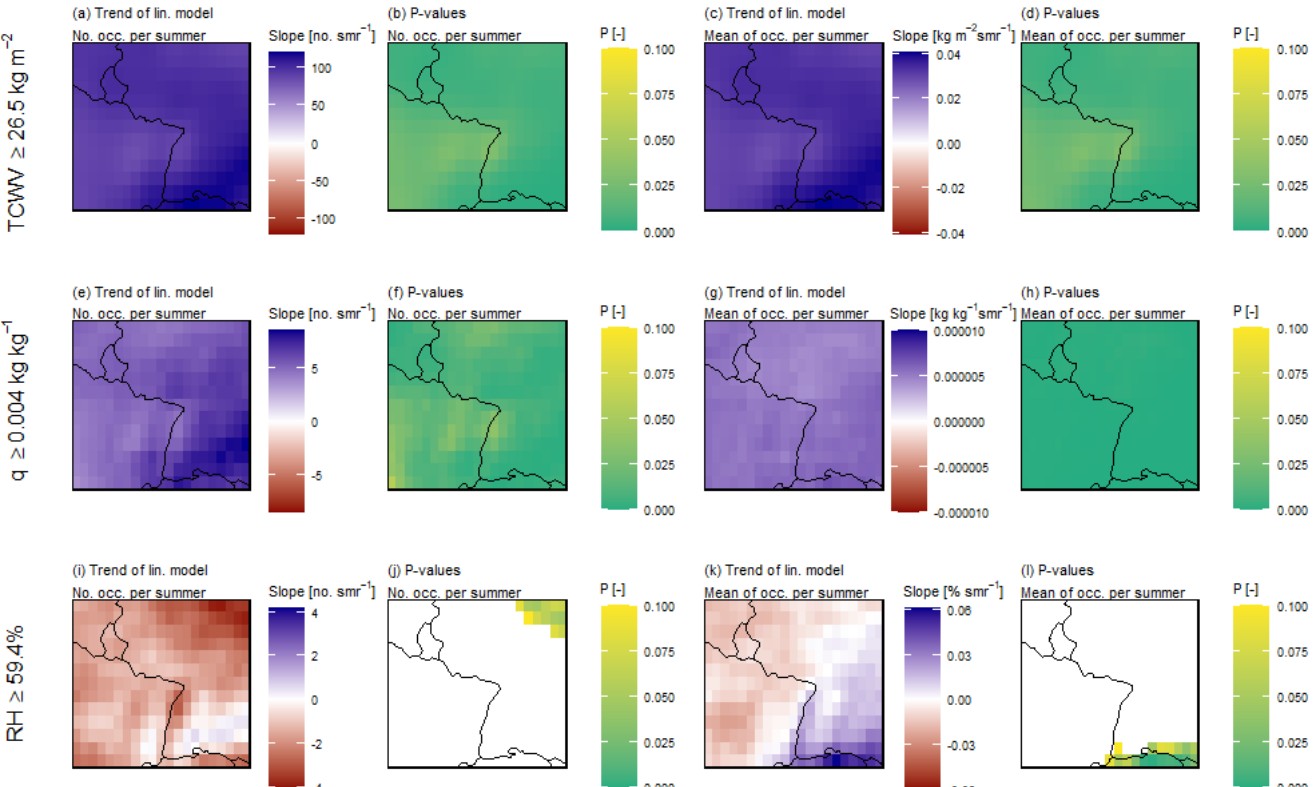

**Figure A2:** Trend analysis of the three variables for moisture (TCWV, q, RH) per summer (smr). The first column (a, e, i) shows the trends of the numbers of hourly occurrences of values above their respective threshold, including their significance-levels in the second column (b, f, j). The third column (c, g, k) shows the trends of the mean values of all hourly occurrences above the threshold and the last column (d, h, l) their respective significance-levels. White areas mark insignificance.

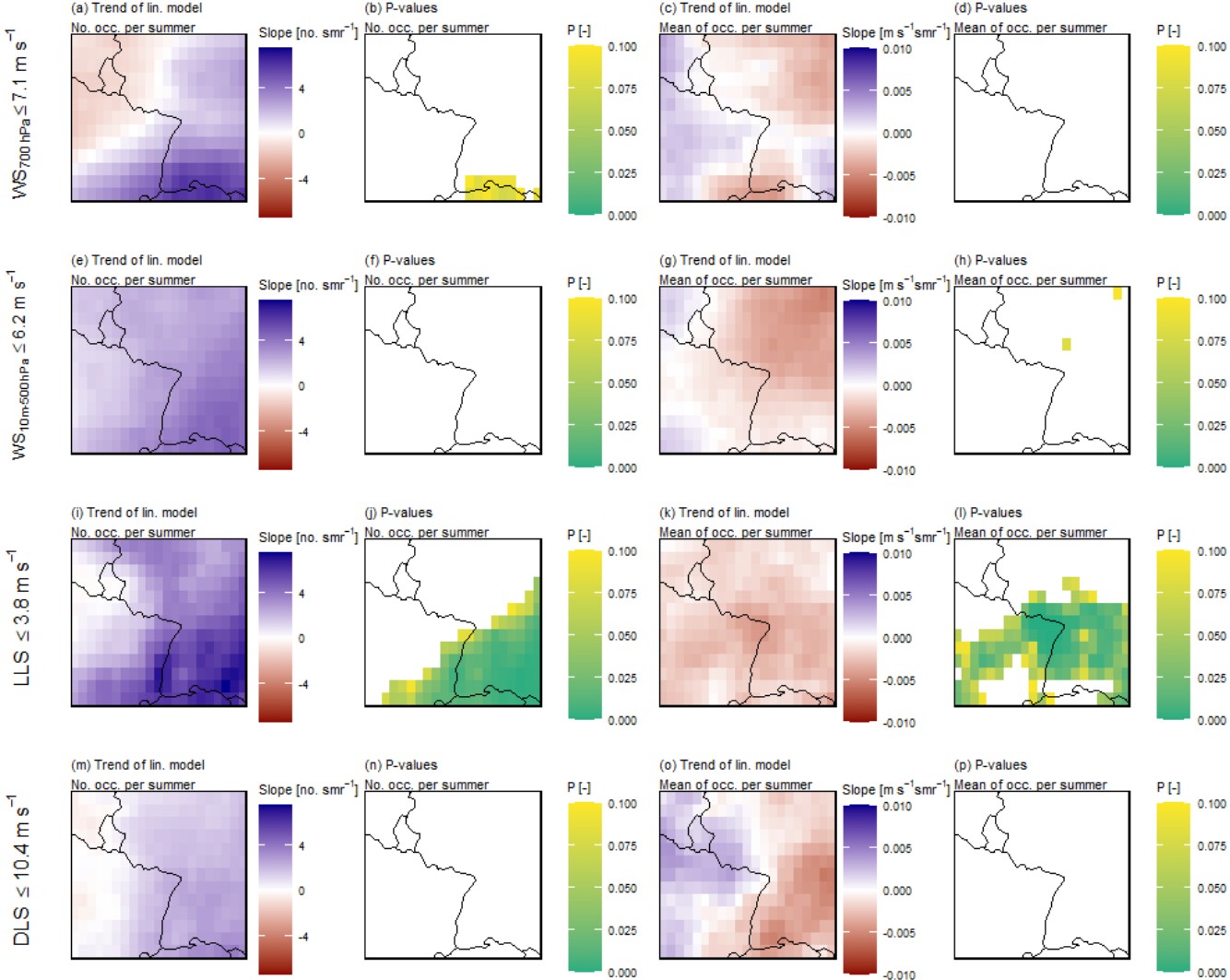

**Figure A3:** Trend analysis of the three variables for storm motion and organisation (WS$_{700\,hPa}$, WS$_{10m-500hPa}$, LLS, DLS) per summer (smr). The first column (a, e, i, m) shows the trends of the numbers of hourly occurrences of values above their respective threshold, including their significance-levels in the second column (b, f, j, n). The third column (c, g, k, o) shows the trends of the mean values of all hourly occurrences above the threshold and the last column (d, h, l, p) their respective significance-levels. White areas mark insignificance.

### Data availability

The flash flood database used in this manuscript is added to the supplements. The RADOLAN radar dataset by the German Weather Service (DWD) is free for download from their open data server [https://opendata.dwd.de/climate_environment/CDC/grids_germany/5_minutes/radolan/reproc/2017_002/, last accessed June

2021]. The ERA5 atmospheric parameters are also free for download from the Climate Data Store (CDS) of the Copernicus Climate Change Service (C3S) [https://cds.climate.copernicus.eu, last accessed November 2021].

**Author contributions**

JM, MN, LP, and LM conceptualized the study. JM collected the flash flood and ERA5 data, carried out the analysis and wrote the first draft of the manuscript. MN provided the processed precipitation radar data. All co-authors (JM, MN, LP, LM, EZ) contributed to and edited the manuscript.

**Competing interests**

The authors declare that they have no conflict of interest.

**Acknowledgements**

This work is supported by the Luxembourg National Research Fund (FNR) via the PRIDE programme for doctoral education (grant PRIDE15/10623093/HYDRO-CSI). We also acknowledge all data source providers for this work: The German Weather Service (DWD) for the RADOLAN radar data, the Copernicus Climate Change Service for the ERA5 dataset as well as Catharin Schäfer and Dr. Hans Göppert from the engineering consultant Wald + Corbe for providing their collection of flash
flood events.

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
