# Peer review of "Atmospheric conditions favouring extreme precipitation and flash floods in temperate regions of Europe"

_Hydrology and Earth System Sciences, 2021_

## Referee Comment (RC2)

Review of: **More frequent flash flood events and extreme precipitation favouring atmospheric conditions in temperate regions of Europe** by Judith Meyer et al.

Ruben Imhoff

Ruben.Imhoff@deltares.nl

January 10, 2022

**Summary**

The authors present an approach to link an increasing flash flood occurrence in central western Europe to a hypothesized increase in extreme precipitation events and favoring atmospheric conditions for these extreme rainfall events using 20 years of DWD RADOLAN data and 40 years of ERA5 reanalysis data. In a three-step approach, the authors show and claim that (based on the presented data) the number of flash floods has increased in the period 1981 − 2020. The same does not hold for the number of extreme rainfall events, but most of the selected favoring atmospheric convective conditions for both event groups show a trend that supports the set hypotheses.

I would like to thank the authors for a well-written and well-structured manuscript that was a pleasure to read. The presented three-step approach to test the hypotheses is easy to follow and is worth keeping. Nevertheless, I still have quite some questions regarding the experimental setup, event selection, underlying data and potential reasons why hypothesis 2 (the increase of extreme rainfall events) is not supported by the data. This would also make it easier to validate whether the methods are sufficient and appropriate to support the conclusions. Although the authors shortly touch upon this in their discussion section, it would put the conclusions in perspective if we have a better idea of the uncertainty of the found trends, based on the available data and choices made in the methods. In the following sections, I will go more into detail on these topics.

**General comments**

**Dataset length**

The authors used RADOLAN data from 2001 − 2020 and ERA5 reanalysis data from 1981 − 2020. I am concerned whether that is a long enough record to make climate-related conclusions? Especially the radar dataset, which only covers 20 years, seems too short to make climate trend-related conclusions. I do see the advantage of the high space-time resolution of radar for such an analysis, and it makes me happy to see it used, but the database length seems not sufficient yet. Although I find it hard to say what the minimum number of years should be in the dataset, I think the work needs at least a more extensive written support for the use of the dataset and the uncertainty that gives in the results.

Regarding the trend found in the data, especially based on the rainfall analysis for the 20 years of RADOLAN data: what does the trend look like if you take out the extreme years 2016 and 2018? I.e., are the trends we see a result of recent extremes?

I wonder if it would make more sense to look back from observed flash floods and extract the ERA5 reanalysis data for these times and locations, instead of partially picking events based on the shorter RADOLAN dataset?

*Flood database*

The first thing I was wondering is how certain the authors are about the increase in the number of reports from 1981 until 2020. Lines 241 – 242 "While barely any events were reported before 2006, two remarkable years are 2016 and 2018, when flash floods occurred particularly often in the study area (23 and 20 occurrences respectively)." Is there a chance that the number of reports also significantly increased over that period? Although I do believe that there is an increase, it may be good to support it by actual discharge time series of the catchments in the study area.

This also directly lead to how the authors have defined a flash flood. This was not directly to clear to me when reading the manuscript. In addition, is a flash flood that occurred on a certain day counted double if it occurred in a different location on the same day? It would be biased to base the frequency of occurrence on such a double counting, while it actually says something about the intensity and spatial extent of the flash flood (and rainfall events). This is also highly relevant, but not the objective of this study.

Concluding, would it be good to take a step back and (1) define what a flash flood is in this study, and (2) search for the events in time that correspond to this definition backed-up by both the literature study and discharge time series? I am aware of the amount of extra work this asks for, but it would make the conclusions stronger.

**Specific comments**

Lines 32 – 33: I would make this sentence a bit longer (to increase readability): E.g. "Flash floods, generally originating from severe convective storm fed by deep moist convection, rank among the most destructive hazards and result in economic losses, damage to infrastructure and high mortality rates (refs)." Or something similar, of course.

Lines 84 – 86 "This generally occurs in case of very weak pressure/geopotential gradients when the mean wind speed and the bulk shear between the surface and the lower to mid troposphere are weak.": True, but what about orography causing or enhancing this?

Lines 88 – 109: I think this paragraph can be shortened. The authors give an extensive overview of proxy parameters used in literature. This is appreciated, but it is, in my opinion, a bit too long and distracting from the main message in the introduction. Perhaps give a couple of examples and then come to the main point of the paragraph.

Lines 116 – 118 "In addition, relative humidity levels decrease at low levels of the atmosphere, connected to rising temperatures, which also reduces the number of thunderstorms (Taszarek et al., 2021a).": Although I am not an expert on this topic, I can image that with higher temperatures evapotranspiration also increases, which leads to higher moisture contents again (besides the fact that the air can contain more moisture at higher temperatures). As said, I am not an expert on this, but I think the statement at least calls for more references.

Line 142 "May to August": Doesn't that leave out some potential late-summer storms in September?

Lines 157 – 159: Can you add some more information about the RADOLAN product? E.g., what kind of radars used, adjusted with rain gauges and how? Hence, how 'good' or reliable is this dataset? Were there any changes in the radar product during the 20 years that also results in different estimations over the years?

Lines 160 – 161 "an extended rain gauge adjustment with supplementary local rain gauges": How many rain gauges were used, what time step was used and what kind of adjustment have the authors applied?

Lines 161 – 162 "We extracted the events for the database from the radar database by identifying 1x1 km grid cells with precipitation amounts ≥ 40 mm h$^{-1}$": But you do not have RADOLAN coverage in the full study area? Or is the study area constrained to the area covered by the RADOLAN observations?

Lines 172 – 174: Is this database giving all the floods for the study domain and which catchments does it contain?

Lines 177 – 178 "The maximum hourly precipitation value was considered the trigger for the flash flood and atmospheric parameters were extracted from the identified grid cell and time.": What about the cells around this grid cell, as their parameters may also have influenced the rainfall that fell there?

Lines 178 – 180: How did you find the flash floods here and the rainfall intensities, as this is outside the RADOLAN data coverage? In addition, do you have time series of the catchments, which could already indicate the presence and timing of a flash flood?

Line 203 "extremely rare in Central Europe": just out of interest (and perhaps worth mentioning), how rare is it (quantified)?

Line 206 "700 hPa": Why have the authors chosen to pick the 700 hPa level?

Line 215 "soil moisture (Swvl) [m3 m-3] at depths of 0-7 cm, 7-28 cm, and 28-100 cm from ERA5": Why have the authors chosen for these three depths and would it make sense to average them in some way, as they will be (cor)related to each other?

Lines 222 – 223 "Therefore, we chose upper or lower boundaries including 75 % of extreme events.": Do the authors mean the events IQR of the extreme events or did I understand it incorrectly?

Line 248 "Between 2001 and 2020, we observed a slight increase in the number of events per year (Figure 3a).": But not a significant one, right?

Lines 266 – 267 "Moisture conditions during extreme precipitation and flash flood events were found to be mostly within the upper percentiles of the overall simulated values.": That is also what you expect seeing the Clausius-Clapeyron (CC) relation and in fact even the 2CC relation for extreme precipitation. It probably deserves mentioning that, including some references (e.g. Lenderink & Van Meijgaard, 2008; Mishra et al., 2012; Manola et al., 2018; Wasko et al., 2018; Dahm et al., 2019).

Lines 269 – 270 "All moisture parameters, and especially RH tend to be even higher during flash flood events compared to general extreme precipitation events (Figure 4d-f).": As clearly not all heavy rainfall events lead to flash flood events, can you also give some event statistics (earlier in the manuscript) between the two groups? What were average rainfall intensities in both groups, does the duration differ, does the size of the rainfall storms differ, etc.? This will give an idea why we see differences between the two groups.

Lines 274 – 277: This also says a lot about the initial catchment wetness prior to a flash flood event. As stated earlier by the authors, the wetter, the quicker a flash flood can occur. Now, from these results, I do not directly see a significant difference between the three groups. Only the 'P' group has somewhat lower initial soil moisture values, which gives the impression that heavy, convective rainfall does more often occur during drier periods. Something which corresponds a bit to the summer weather patterns in Northwest Europe. It also suggests that initial soil moisture conditions were on average not different from other days in the studied periods, so the flash floods are mostly a result of the weather system and not initial conditions here.

In addition, perhaps it is interesting to show the soil moisture as a relative scale (so % of the capacity).

Line 291 "These findings were particularly significant for the northern part of the study area (Figure 5b).": Any idea why in the north?

Lines 332 – 335: How is the trend if you take out 2016 and 2018?

Lines 361 – 362 "Regarding low wind speeds and weak bulk shear, we found slightly increasing but barely significant trends.": But you did find a significant trend for LLS, right?

Lines 380 – 382: This might also be related to the finite gauge-adjusted radar dataset of 20 years.

Lines 407 – 409 "Future analyses could incorporate the intra-annual temporal distribution of extreme precipitation events. Perhaps, formerly evenly distributed rainfall events tend to occur more condensed within a few days.": This is something you could already focus on in this study, by also looking at longer event durations. So what if you don't only look at 1-h accumulations, but also 6-h, 24-h, etc?

Lines 420 – 422 "In addition to the hypothesis, we found mostly higher upper (0-7 cm) and lower (28-100 cm) layer soil moisture during flash flood events compared to general extreme precipitation events.": This did not seem that significant in the results.

Figure 1c: I suggest to put here an actual DEM with a higher resolution, which makes the mountain ranges and the differences between them clearer.

Figure 4: Would the differences (which are clearly present!) become clearer when you take the P and FF events out of the all group?

Figure 5:

1. An idea for the figure, make the color scale discrete instead of continuous, then it is easier to distinguish the actual values.
2. In addition, the slope is in [unit] per year. So, don't forget to give the unit.
3. To get an idea of the timeseries underneath, could the authors provide for one pixel the timeseries + trend?

**Technical corrections**

Line 25: We tend not to confirm a hypothesis, but it is supported by the results.

Line 112 "point out, that CAPE": the comma can be removed.

Lines 240 – 241 "0.382 events per year (p-value 0.039)": These values are not exactly the same as in Fig. 2a.

Line 247 "precipitation sums >40 mm h-1": you can remove "h-1", as it states precipitation sums and not intensities.

Line 404 and 418 "confirmed": supported by the found data.

Conclusions section: as many people only quickly read through the abstract and conclusions, make sure to write out abbreviations (once again) in the conclusions.

Figure 3c: "intensities" in y label should be "sums".

Figure A1: K-Index increase is also clearly significant in the north. Perhaps worth mentioning in the results.

**References**

Dahm, R., Bhardwaj, A., Sperna Weiland, F., Corzo, G. & Bouwer, L.M. (2019). A Temperature-Scaling Approach for Projecting Changes in Short Duration Rainfall Extremes from GCM Data. Water, 11, 313. https://doi.org/10.3390/w11020313.

Lenderink, G. & Van Meijgaard, E (2008). Increase in hourly precipitation extremes beyond expectations from temperature changes. Nat. Geosci., 1, 511–514. https://doi.org/10.1038/ngeo262.

Mishra, V., Wallace, J.M. & Lettenmaier, D.P. (2012). Relationship between hourly extreme precipitation and local air temperature in the United States. Geophysical Research Letters, 39, L16403. https://doi.org/10.1029/2012GL052790.

Manola, I., van den Hurk, B., De Moel, H. & Aerts, J.C.J.H. (2018). Future extreme precipitation intensities based on a historic event, Hydrol. Earth Syst. Sci., 22, 3777–3788. https://doi.org/10.5194/hess-22-3777-2018.

Wasko, C., Lu, W.T. & Mehrotra, R (2018). Relationship of extreme precipitation, dry-bulb temperature, and dew point temperature across Australia. Environ. Res. Lett., 13, 074031. https://doi.org/10.1088/1748-9326/aad135.

---

## Referee Comment (RC3)

Meyer et al. 2021, "More frequent flash flood events and extreme precipitation favouring atmospheric conditions in temperate regions of Europe"

**Summary:** This study investigates changes in the frequency of flash flood events in central Europe and concomitant changes in extreme precipitation and atmospheric variables promoting heavy rainfall. The database of flash flood events is compiled from reinsurance, literature review, and personal communication and connected with heavy rainfall events using a radar network across Germany to determine ERA5 grid cells in which the radar indicated rainfall exceeds 40 mm h$^{-1}$. ERA5 grid cells containing and neighboring the precipitation event are then used to study changes in atmospheric variables promoting heavy rainfall using instability indices, moisture, and storm motion values. While the number of flash flood events show a slight increase over time, precipitation events and their rainfall characteristics exhibit little change. These precipitation events are often associated with some CAPE, high moisture, and weak winds. Of these variables, moisture shows the largest and most significant increase over the study period, while CAPE and CIN showed a loss noticeable increase, and storm motion/shear in some places exhibits decreases. Therefore, while the authors showed that flash flood events are increasing in frequency over time, they could not link this to changes in rainfall, but rather more favorable rainfall environments. This is an interesting study, but several major points need to be addressed before publication.

**General comments:**
- A major limitation of this study is the lack of flash flood events, particularly before 2006, and how they are identified. While the authors acknowledge this limitation in lines 332-333, I wonder if this is not an issue of a lack of flash flood events in the past, but a limitation of the observational record they use to define flash flood events. Flash floods are defined by news reports, prior literature (I am guessing case studies?), water agency reports, and reinsurance data, which are all prone to human error, including the need for people to observe the flood and report it as noteworthy. I wonder if they can incorporate any physically based indications of a flood event by including streamflow observations. This could address the dearth of floods prior to 2006 and remove some of the inherent bias in the human-based indications of flooding.
- The connection between the flash flood events and extreme rainfall is unclear in the present form of the manuscript. The only description of how these events is linked are in lines 172–173 where the authors state that they include all floods that directly follow extreme precipitation. What is the temporal scale used to determine what "directly follows" means? Is this an hour, several hours, or a day? Please be explicit in stating this. Also, what is the spatial requirement for a flood event being connected to an extreme precipitation event? Please describe this in more detail. Finally, the independence of flood and precipitation events must be discussed. For example, if a flood event occurs on two consecutive days, is that considered the same event? Again, please discuss this in more detail.
- It seems odd to me that despite an increase in more favorable rainfall environments over time, little trend is observed in changes to extreme rainfall. While the authors discuss

possible reasons why this is in the discussion section, I wonder if it would be helpful to include other sources of rainfall data (like rain gauges or even ERA5, despite their limitations), to see if the same lack of a trend is reproduced.

**Specific comments:**
- Lines 59–62: There are some contradictions in these lines as to how you refer to precipitation events that trigger floods. In line 59, you state that they are characterized by high rainfall amounts over a short period of time, while in the next few lines, you say that the rainfall also lasts over longer periods of time. What do you mean here? Please be clear if these are short or long duration rainfall events. Perhaps providing typical durations could be helpful here.
- Lines 67–70: I am not sure I understand what you are saying here–which processes are you referring to? Do you mean the upscale growth of convective cells into organized convection, like a mesoscale convection system? Please be more specific.
- Line 80–81: I would also cite Schroeder et al. (2016) regarding a larger warm cloud depth leading to higher precipitation efficiency.
    - Schroeder, A., et al., 2016: Insights into atmospheric contributors to urban flash flooding across the United States using an analysis of rawinsonde data and associated calculated parameters. *J. Appl. Met. and Clim.*, **55**. Doi: https://doi.org/10.1175/JAMC-D-14-0232.1
- Lines 83–84: Large rainfall systems can also result in long duration storms (Doswell et al. 1996).
- Line 88: I would start a new paragraph here at "Proxy parameters.." since this paragraph is already quite long.
- Lines 91–93: The results of this study seem to contradict your previous sentence stating that bulk wind shear can be used to estimate precipitation efficiency if heavy precipitation occurs over a variety of DSL values. How do you reconcile this conflict?
- Line 114: Please also cite Rasmussen et al. (2017), as they were among the first to discover the increasing CAPE/decreasing CIN paradigm:
    - Rasmussen, K. L., A. F. Prein, R. M. Rasmussen, K. Ikeda, and C. Liu, 2017: Changes in the convective population and thermodynamic environments in convection-permitting regional climate simulations over the United States. Climate Dyn., 55, 383–408, https://doi.org/10.1007/S00382-017-4000-7.
- Line 175–176: Are this 8 neighboring grid cells centered around the precipitation event grid cell? What if the precipitation event takes up multiple grid cells?
- Lines 205–215: Did you perform a sensitivity test to see if taking the RH or winds at different pressure levels aside from 700 hPa affected your results?
- Line 240: how would your results look if you omitted the year 2016? Would it still be an increasing trend?
- Line 258: what does "all hourly values" refer to? Is it the time of the precipitation event and the flood event combined or something else? This needs to be described in more detail. Also, how long, on average, are the precipitation events and the flood events?

- Lines 261–262: Half of the distribution is above 100 J Kg$^{-1}$ of CIN, which is moderate CIN, so I do not believe saying high values of CAPE are often accompanied by low values of CIN is entirely accurate.
- Lines 273–274: Given the low wind speed and weak DSL, this likely indicates that these storms are slow-moving single-cell thunderstorms. This is interesting, because many flood-producing storms tend to be larger and more organized mesoscale convective systems (Ashley and Ashley 2008; Schumacher and Johnson 2006):
    - Ashley, S. T., and W. S. Ashley, 2008a: The storm morphology of deadly flooding events in the United States. Int. J. Climatol., 28, 493–503, https://doi.org/10.1002/joc.1554.
    - Schumacher, R. S., and R. H. Johnson, 2006: Characteristics of U.S. extreme rain events during 1999–2003. Wea. Forecasting, 21, 69–85, https://doi.org/10.1175/WAF900.1.
- Line 282: CAPE at or exceeding 100 J kg$^{-1}$ is not high, but rather weak. I recommend you edit language throughout the paper to reflect this.

**Technical corrections:**
- Line 66: I would consider using a different word rather than "neglected" such as "slowed" or "halted".
- Line 228: I would replace "prove" with "test".
- Figure 3: Please describe what each of the panels are showing.
- Line 281: I believe you meant to stay Table 2 instead of Table 1, correct?

---

## Author Comment (AC1)

**Authors' response to Reviewer 1**

**[hess-2021-628-RC1]**

*We thank the reviewer for taking the time for the evaluation of our manuscript (hess-2021-628). Below we address the reviewer's comments (full text) indented by arrows and coloured in blue.*

Their dataset and methods adopted in analysis are seriously flawed. The three hypotheses that they raised in the manuscript cannot be validated based on the existing analytical framework.

→ *From the comments made by the referees we understand that there is a need for clarification on both the definition of flash floods and the way how the related database has been built. Here below, we will provide additional elements that shall eventually also be included in the revised version of the manuscript.*

Aside from the technical issues, a key problem is that throughout the manuscript the authors do not specially define what is exactly a "flash flood" (in their perspective). We all know flash floods can be different from other types of riverine floods in various ways. However, it is never proper to simply classify floods during the summer months as flash floods (as distinguished from the winter floods). Without clarification of the basic concept, some of the sentences seem logistically biased. For instance, "The development of flash floods relies on long-lasting, extreme precipitation" (Line 108). This is not true, since extreme rainfall does not have to be "long-lasting" to generate a flash flood, although it is true for a subset of flash floods (not vice versa).

→ *We agree that multiple flash flood definitions have been proposed in literature. A non-exhaustive list of references defines flash floods as:*
   - *A subset of pluvial floods (Owen et al., 2018),*
     - *exhibiting different characters than river floods (WMO, 2017)*
     - *runoff rates often exceeding by far those of other flood types, due to the rapid response of a catchment to intense rainfall (Borga et al., 2010)*
     - *often occurring on steep slopes (Van Campenhout et al., 2015)*
     - *with high flow velocities (Van Campenhout et al., 2015)*
     - *composed of less than 30% of solid material (Kron, 2011)*
   - *Being caused by intense rainfall (Owen et al., 2018),*
     - *with event durations < 34 h (Marchi et al., 2010)*
     - *short-lived storms with high intensities < 24h (Gaume et al., 2009)*
     - *high intensity rainfall, mainly of convective origin and affecting small areas (Borga et al., 2010)*
     - *rainfall totals with return periods exceeding 50 yrs (Marchi et al., 2010)*
     - *rainfall totals exceeding 100 mm over a few hours (Gaume et al., 2009)*
     - *mainly of convective origin, spatially confined and often orographically enhanced (Gaume et al., 2009)*
   - *Being characterised by short response times (Marchi et al., 2010)*
     - *generally, less than 6h (Marchi et al., 2010)*
     - *with rapidly rising and falling limbs of the hydrograph (Owen et al., 2018)*
     - *occurring within minutes to several hours, depending on the region (WMO, 2017)*
   - *Remaining very difficult to forecast (WMO, 2017; Owen et al., 2018)*

- - - - - - - - - - - - - - - - - - - - - - - - - - - - - - - - - - - - - - - - - - - - - - - - - - - - - - - - - - -
- ▪ *Flash floods developing at spatial and temporal scales that conventional hydro-meteorological observation systems are not able to monitor (HYDRATE, 2008)*
  - o *Occurring in rather small catchments (WMO, 2017; Owen et al., 2018)*
    - ▪ *< 1000 km² (Marchi et al., 2010)*
  - o *< 500 km² - varying from tenths to a few hundreds of km² (Gaume et al., 2009)*
- → *The flash flood definition criteria used in the references listed above consist mainly of metrics referring to the intensity of events observed via hydro-meteorological monitoring networks. When occurring in areas not covered by a monitoring instrument, extreme rainfall-runoff events may eventually remain undetected with this approach. Therefore, we propose here an alternative method - tapping into multiple sources of extreme hydro-meteorological events, combining scientific papers, agency reports, insurance inventories, personal communications, and newspapers. Moreover, we connected the flash flood database to the RADOLAN data matching it with the onset of a precipitation exceeding 40 mm/h (as defined by the DWD, leading to extreme weather warnings) within the grid cell of the flash flood. We will further improve this approach by considering a wider area around the occurrence location for identifying the corresponding precipitation event. The precipitation database should account for unnoticed flash floods.*
- → *We eventually relied on this multi-source approach for: (i) identifying extreme convective precipitation events in summer, triggering floods with considerable stream power, erosive force and impact potential to (inundated) infrastructures (line 172f); (ii) accounting for the high spatial heterogeneity that characterizes extreme hydro-meteorological events during the summer season - as opposed to inundations occurring in large river floodplains, caused by advective precipitation events (albeit not exclusively) - mainly during winter months.*
- → *Note that the vast majority of events identified through our query remained bound to catchments smaller than 120 km$^2$. Large summer floods, triggered by prolonged rainfall over extended areas and that may have occurred on larger rivers (e.g., Moselle, Rhine), were disregarded in our study.*
- → *The table here below relates to (i) the information sources used in our study, (ii) the criteria retained for identifying flash flood events, and (iii) the number of events that had eventually been identified. Note that the time period covered by our multi-source query spans from 1981 to 2020.*

| | Criteria | No. from this source |
|---|---|---|
| *Scientific papers* | *keywords: "flash flood"* | *4* |
| *Reports (LFU, LUA, Ministère de l'Ecologie, du Développement durable et de l'Energie)* | *keywords: "Sturzflut", "Hochwasser", "crue éclair", "crue subite", "inondation", during summer months, excluding big rivers (Moselle, Rhine)* | *Göppert: 21* *Johst: 7* *Pfister: 4* *Ministère de l'Ecologie, du Développement durable et de l'Energie: 1* |
| *Insurance reports* | *keywords: "inondations" in combination with thunderstorms, heavy/extreme precipitation, summer.* *All floods listed have cause mentionable damage* | *1 (insurances only report major events, i.e. Braunsbach)* |
| *Personal communication* | *local hydrological events caused by heavy convective rainfall that have led to damage and made it to the awareness of the collector* | *23* |
| *Newspaper* | *keywords: "Sturzfluten", "Inondations", in summer, after thunderstorm/ intense rainfall* | *France 3: 16* *Trier: 1* |

| DWD Radolan data | Precipitation intensity per grid cell exceeding 40mm/hour → We will have a closer look in the revised version, to be able to better identify P for FF, that should have been covered by RADOLAN. | In 45 (of 83) cases a P event could be related to a FF event. 5 were before the start of the RADOLAN, and < 9 outside the area covered by RADOLAN. |
|---|---|---|

→ *As shown by the manifold definitions provided in literature, a precise and clear definition of flash floods remains challenging. For our study, we considered extreme pluvial floods, as reported in scientific papers, agency & insurance reports, personal communications, and newspapers. Note that based on the available RADOLAN weather radar dataset we could eventually attempt to quantify the precipitation amount, intensity, and duration for some of the reported events in a backward approach. However, this would not improve the quality of the database required for our study – targeting in the first place the building of a comprehensive set of extreme (summer) rainfall-runoff events. The response time between the precipitation peak and the runoff peak occurred within only a few hours for each event. Since both precipitation and discharge data were only available for a subset of events, we opted for building a database reporting on the sole occurrence of extreme summer rainfall-runoff events (as per the criteria listed above). Note that the catchments in which the retained events occurred were all "small" - spanning from individual slopes (where major surface runoff had been reported) to catchments up to the size of a bit more than 120 km² (e.g., Ernz Blanche river, Starzel river). For all reported events, stream power and inundated water levels were strong enough to create substantial impact and damage (e.g., the displacement of large objects, such as cars).*

→ *We will revise the introduction of our manuscript paragraphs to better reflect a) on the challenges inherent to the definition of flash floods, as well as on b) the criteria that we have eventually retained for selecting events for our summer extreme rainfall-runoff event database. Basically, we are looking at extreme, pluvial, small-scale floods with a high impact that we call flash floods.*

Another concern of mine is that the flash flood database is not consistent in space and time. Any trend analysis based on the dataset would not be able to generate true insights into the real world. The authors also admit that the database is non-exhaustive. I would suggest the authors to demonstrate their efforts in making the database at least consistent in time. Otherwise, people would argue whether the significant trend is due to sampling biases or not. This corresponds to their first hypothesis (Line 404-405).

→ *We admit that we cannot guarantee an equally consistent database for the entire time period. While CCR has inundation data available online since 1989, Wald + Corbe collected data more systematically since 2006 and France 3 only since 2012. Moreover, we have checked the database by the European Severe Storm Lab for heavy precipitation, that also shows biases in space and time. As long as holding on to the database creation based on reports, this bias will inherently remain. The advantage of this approach is, however, to obtain a sample of extreme pluvial floods much larger, as it can be inferred from gauge catchments alone.*

→ *The cleanest option we can suggest is to take back this first part of the hypothesis and focus on the identification of their atmospheric parameters and calculate the trend based on them as a proxy.*

In addition, the authors use cumulative statistics to quantify the occurrences of flash floods for each year. Since floods cluster in space and time, the authors need to be aware of the issue of repeated counting. This is relevant to their second hypothesis where they evaluate trend in the occurrences of extreme rainfall. It would be biased to count the number of grids with rain rate exceeding certain thresholds. The statistics thus reflect the combined effect of intensity and spatial coverage of rainfall, not changes in the frequency.

→ *We agree that clustering can be an issue, due to the dependency of floods on event and pre-event conditions. Extreme events that are highly variable in time and space, such as flash floods and hailstorms, are telling examples in this respect - as shown for example by Changnon (1984) for hailstorms in the American Midwest. The flash flood clustering effect is also shown in Figure 2b of our study - showing the temporal occurrence of floods. We counted every single event, also when they occurred in the vicinity of another flood, or within a few consecutive days. The connecting element of flash floods are the related meso-scale atmospheric systems and/or similar pre-event conditions. In 2016 and 2018, for example, most flash flood events happened within 2 weeks across our study area. Multiple events may be linked (e.g. Piper et al, 2016), but they eventually also cause more damage and fatalities than a single flash flood event that may have occurred in only one isolated catchment. Therefore, we argue that counting all flash floods is important information, as multiple events should be weighted stronger than isolated events.*

→ *What was pointed out by the reviewer, is that "The statistics thus reflect the combined effect of intensity and spatial coverage of rainfall, not changes in the frequency." This is an important part of what we try to show. We had already aggregated the small radar grid cells to the resolution of ERA5 grid cells. This was to extract the atmospheric parameters in the next step. In the revised version we offer to cluster the radar grid cells by events for the second sub-hypothesis and count the number of grid cells contributing to an event, as we do not want to lose the spatial information and extent of the data. If high intensities of precipitation are present over a larger spatial area, the chance of a flash flood to occur should increase. We moreover suggest adjusting the wording and move from "more frequent precipitation events" to a "more frequent occurrence of precipitation intensities potentially generating flash floods".*

*Changnon, S., 1984: Temporal and spatial variations in hail in the upper Great Plains and Midwest. J. Climate Appl. Meteor., 23, 1531-1541.*

Lastly, I did not see significant increases in the proxy parameters for flash flood potential. This is mainly a concern with Fig. 5. Increases in moisture content are kind of expected according to the Clausius-Clapeyron relationships, but other than that, the other two proxy parameters show negligible significance (especially for DLS). In addition, flash floods are tied to comprehensive combinations of atmospheric conditions. By examining the trend in individual component of the comprehensive conditions as the authors did here offer limited insights into the real changes in flood potential. The threshold values are also chosen in a subjective way that needs further justification.

→ *It is true that many factors are involved in the development of a flash flood, which also need to interact in the right proportions. If individual parameters (or indices) would adequately describe their formation, then flash flood forecasts would most certainly already be routinely done. However, it is not our aim to comprehensively describe and mechanistically relate the occurrence of flash floods to individual parameters and/or indices of atmospheric conditions. We will clarify this aspect in our manuscript.*

→ *We confirm that the trends identified for DLS and CAPE are weak and largely insignificant. This was reported in the manuscript (for DLS, line 297-300 & line 413-415 / for CAPE, line 289-292). We will further develop on this finding in the revised version of our manuscript. The expectation of the Clausius-Clapeyron relationship is a 7%/K scaling in water vapour pressure if relative humidity remains invariant under climate change. While it is a frequent assumption in most climate change scenarios, it is after all an assumption.*

→ *In order to account for the combined occurrence of flash flood relevant atmospheric parameters, we added the subchapter 3.5, Fig. 6. This simple approach using low thresholds already excludes some of the occurred events. Any effort to specify this would not do justice to the variety of extreme rainfall events. Moreover, we considered using a GLM, but rejected the idea, as the parameters are not independent from one another.*

→ *The thresholds were originally chosen in a range relevant in literature. These values seem however often too high to include most detected events. Therefore, we adjusted the approach, while lowering the thresholds to include 75% of all our identified precipitation events. These choices are described*

*in line 221-224 and 281-284, and a discussion of it is added in line 369-376. We will develop this aspect in more detail in the revised version of the manuscript.*

I would not go into any further details about the presentation of the manuscript. Some of the sections (like Introduction, Discussion) need to be shortened and merged. These issues are relatively less important compared to the aforementioned concerns of mine.

→ *We had a close look at our manuscript again and also see the need for shortening some sections of the manuscript. We will take this into account in the revised version, whereby we - as previously stated - will also develop some aspects more clearly and reduce the ambiguities that may have prevailed in some statements of the initial version of our manuscript. However, we do consider a certain level of detail necessary, since this manuscript covers an interdisciplinary range of topics (flash floods, extreme precipitation, and atmospheric conditions) and that it might be read by specialists with one or the other background. The long introduction was initially meant to give valuable context about both the meteorological and hydrological aspects dealt with in the manuscript.*

---

## Author Comment (AC2)

**Authors' response to Reviewer 2**

**[hess-2021-628-RC2]**

We thank the reviewer Ruben Imhoff for his evaluation of our manuscript and his many helpful comments (hess-2021-628). Below we address the reviewer's comments (full text) indented by arrows and coloured in blue. We appreciate the efforts by the reviewer, which will help to improve our manuscript.

**General comments**

**Dataset length**

The authors used RADOLAN data from 2001 – 2020 and ERA5 reanalysis data from 1981 – 2020. I am concerned whether that is a long enough record to make climate-related conclusions? Especially the radar dataset, which only covers 20 years, seems too short to make climate trend-related conclusions. I do see the advantage of the high space-time resolution of radar for such an analysis, and it makes me happy to see it used, but the database length seems not sufficient yet. Although I find it hard to say what the minimum number of years should be in the dataset, I think the work needs at least a more extensive written support for the use of the dataset and the uncertainty that gives in the results.

→ *This is a valid point. Prior to these analyses we analysed 98 precipitation stations with data for the time period 1954-2018. For this long time period, only daily precipitation amounts are available consistently. We analysed the daily precipitation maxima, as well as all days with precipitation amounts higher than 50 mm/day. With these analyses we were not able to detect significant trends either. Parts of the analyses were published in a conference poster at EGU (Meyer et al., 2020). Our conclusion was that daily data is insufficient for thunderstorm events, even though the daily precipitation sum should reliably indicate extreme precipitation events. However, we believe we missed many thunderstorm cells within the coarse network of the stations. As both, the long term coarsely resolved dataset as well as the highly resolved, short-term dataset presented in this manuscript, show the same results, we considered the hypothesis II (increase in precipitation) as rejected for the analysed time period. We will extend the discussion by one section about the data base length and the previous findings of the daily station data.*

→ *Meyer, J., Douinot, A., Zehe, E., Tamez-Meléndez, C., Francis, O., and Pfister, L.: Impact of Atmospheric Circulation on Flooding Occurrence and Type in Luxembourg (Central Western Europe), EGU General Assembly 2020, Online, 4–8 May 2020, EGU2020-13953, https://doi.org/10.5194/egusphere-egu2020-13953, 2020*

Regarding the trend found in the data, especially based on the rainfall analysis for the 20 years of RADOLAN data: What does the trend look like if you take out the extreme years 2016 and 2018? I.e., are the trends we see a result of recent extremes?

→ *Regarding the RADOLAN dataset, we did not find a trend. In the sum, 2016 does also not show unusually high values. Maybe you meant 2006? See below the graphs following the experiments. Especially leaving out the many threshold exceedances in 2006 shows the originally hypothesised trend in extreme precipitation events. We will discuss the influence of these extreme years more thoroughly.*

[Figure]

*Figure 1: Number of RADOLAN grid cells with hourly precipitation intensities exceeding the threshold of 40 mm/h. Top plot: including the entire time series, middle plot: excluding the two years, 2016 and 2018, third plot, excluding the two most extreme years 2006 and 2018.*

I wonder if it would make more sense to look back from observed flash floods and extract the ERA5 reanalysis data for these times and locations, instead of partially picking events based on the shorter RADOLAN dataset?

→ *This was indeed an option. The atmospheric parameters relevant for the extreme precipitation and subsequent flash floods should, however, be "most characteristic" directly before the onset of the rainfall event. That is why we chose the time just before the corresponding precipitation event. Flash floods might occur a few hours after the onset and peak of the precipitation event. Moreover, the actual times and locations of the flash flood occurrences were hard to determine, as the sources*

*of our database (e.g. reinsurance, newspaper) name mainly damages or event descriptions, rather than hydrologically relevant data. Therefore, we identified a high intensity rainfall event in the ERA5 grid cell of the flash flood to extract a time. As in some cases this approach did not lead to matches, we will extend this approach to search for identified rainfall events the neighbouring ERA5 grid cells as well and lower the threshold where necessary, to be sure to identify a rainfall event for each flash flood within the spatial and temporal extent of the RADOLAN data. For the flash floods that occurred outside the spatial or temporal range of the RADOLAN data, we extracted ERA5 data during the evening (6 pm) of the day and location of the flash flood independent of the RADOLAN data.*

**Flood data base**

The first thing I was wondering is how certain the authors are about the increase in the number of reports from 1981 until 2020. Lines 241 – 242 "While barely any events were reported before 2006, two remarkable years are 2016 and 2018, when flash floods occurred particularly often in the study area (23 and 20 occurrences respectively)." Is there a chance that the number of reports also significantly increased over that period? Although I do believe that there is an increase, it may be good to support it by actual discharge time series of the catchments in the study area.

→ *Before the choice of using a database that was collected through various sources, we analysed discharge data in the region (entire Moselle catchment). Therefore, we collected data for time series as long as possible. Long times series are, however, mainly available for large rivers, such as the Moselle or other bigger stream gauges, but not for catchments, in which flash floods occur. Moreover, data is often only available on a daily resolution. We have conducted several analyses of specific discharge using 79 stations within the region with catchments < 300 km² and found it hard to extract flash floods or high floods from these data. High flows in the past (1980s) were often caused by zonal precipitation in the Vosges mountains. Some regional flash floods that were of major importance and that we know well (i.e. Ernz Blanche 2016 & 2018), were to some extent detected by discharge data, but the overall time series are too short for any long-term analysis. Other events were so small and even outside streams, that they were not even captured by any stream gauge. We concluded that the inconsistencies in this type of streamflow-based dataset would be even bigger than the one presented in the manuscript. Apart from actual flash floods we have also made analyses about the number of scientific reports on the topic, which also started to increase around that time period (beginning 2000), when the topic received more attention. While a better database would be desirable, flash floods rely on site inspections.*

→ *The cleanest way to think of, is taking back the first sub-hypothesis about the increase in flash floods. While we do believe, that there was an increase, the database cannot be independent and consistent enough to clearly answer this hypothesis.*

This also directly leads to how the authors have defined a flash flood. This was not directly to clear to me when reading the manuscript. In addition, is a flash flood that occurred on a certain day counted double if it occurred in a different location on the same day? It would be biased to base the frequency of occurrence on such a double counting, while it actually says something about the intensity and spatial extent of the flash flood (and rainfall events). This is also highly relevant, but not the objective of this study.

→ *We will try to sharpen the definition of flash floods, as stated in detail in the response to RC1.*

→ *Flash floods were indeed counted twice if two occurred on the same day in neighbouring catchments. While the meso-scale atmospheric situation might be the same, the floods*

*develop independently from one another. We therefore find it valid to count each flood separately.*

Concluding, would it be good to take a step back and (1) define what a flash flood is in this study, and (2) search for the events in time that correspond to this definition backed-up by both the literature study and discharge time series? I am aware of the amount of extra work this asks for, but it would make the conclusions stronger.

→ *Regarding point (1), we will work on a clearer definition according to the details stated in the reply to the other reviewers. Regarding point (2), this would be extremely challenging, as described above. Highly resolved data would catch quickly rising floods, but these are only available for a short period of time and for a few stations of larger rivers (last two decades). Therefore, most floods occur unmeasured.*

**Specific comments:**

Lines 32 – 33: I would make this sentence a bit longer (to increase readability): E.g. "Flash floods, generally originating from severe convective storm fed by deep moist convection, rank among the most destructive hazards and result in economic losses, damage to infrastructure and high mortality rates (refs)." Or something similar, of course.

→ *Thank you, we will revise this.*

Lines 84 – 86 "This generally occurs in case of very weak pressure/geopotential gradients when the mean wind speed and the bulk shear between the surface and the lower to mid troposphere are weak.": True, but what about orography enhancing this?

→ *Indeed, orography definitely plays a role in modifying the near-surface wind field (convergence zones), which often leads to the initiation of storms near (low) mountain ranges (http://www.eumetrain.org/satmanu/CMs/ConOro/print.htm). Many events around Luxembourg are also connected to the surrounding mountainous areas of the Moselle valley, or even guided by the orographic transition from Gutland to Ösling, as Schmithüsen wrote in "Das Luxemburger Land" in 1940. We will add a sentence about this.*

Lines 88 – 109: I think this paragraph can be shortened. The authors give an extensive overview of proxy parameters used in literature. This is appreciated, but it is, in my opinion, a bit too long and distracting from the main message in the introduction. Perhaps give a couple of examples and then come to the main point of the paragraph.

→ *Okay, we will consider shortening it, even though we think that it is important to get a feeling for realistic ranges of the parameters.*

Lines 116 – 118 "In addition, relative humidity levels decrease at low levels of the atmosphere, connected to rising temperatures, which also reduces the number of thunderstorms (Taszarek et al., 2021a).": Although I am not an expert on this topic, I can image that with higher temperatures evapotranspiration also increases, which leads to higher moisture contents again (besides the fact that the air can contain more moisture at higher temperatures). As said, I am not an expert on this, but I think the statement at least calls for more references.

→ *This is indeed expressed a bit unfortunate and we will clarify this statement and add extra references here. Unfavourable environments for the initiation of deep moist convection seem to*

*have increased according to Taszarek et al., 2021 a. This is despite an increase in instability (CAPE), as the convective inhibition (CIN) seemed to increase at the same rate. Leopore et al., 2021 and our study found equivalent results. The hindrance of the vertical rearrangements is probably the limiting factor according to the above-mentioned studies. Of course, the saturation vapor pressure increases with increasing temperature, and there is an increase of low-level moisture. Yet, especially on hot summer days, when soils are dry, there are water limitations for evapotranspiration. Overall, the rising air temperatures seem to outweigh the increase in specific moisture and the relative humidity is after all lower. We will check some further literature about evaporation.*

→ *Lepore, C., Abernathey, R., Henderson, N., Allen, J. T., & Tippett, M. K. (2021). Future global convective environments in CMIP6 models. Earth's Future, 9, e2021EF002277. https://doi.org/10.1029/2021EF002277*

Line 142 "May to August": Doesn't that leave out some potential late-summer storms in September?

→ *It is leaving out one flood event that occurred in the very beginning of September. From a climatological perspective, most thunderstorms occur during the months May-August (Flohn, 2954, Weischet & Endlicher, 2000). In September, the weather is a lot calmer. Therefore, we believe that the omission of this month will not change the study's results.*

→ *Flohn H. 1954: Witterung und Klima in Mitteleuropa, 2. Auflage. Forschungen zur Deutschen Landeskunde, 78, S. Hirzel Verlag, Stuttgart, S. 214.*
*Weischet, W. & W. Endlicher. 2000. Regionale Klimatologie. Teil 2. Die Alte Welt. Teubner.Stuttgart, Leipzig. 625 pp.*

Lines 157 – 159: Can you add some more information about the RADOLAN product? E.g., what kind of radars used, adjusted with rain gauges and how? Hence, how 'good' or reliable is this dataset? Were there any changes in the radar product during the 20 years that also results in different estimations over the years?

→ *Good point. Thanks. A more detailed description of the underlying radar data set is required in any case. We'll catch up on this. The underlying dataset is the Radar-based Precipitation Climatology Version 2017.002 (Winterrath et al., 2018), which in turn is based on the standard RADOLAN product. The RADOLAN method is however a real-time application. It uses an 'online' rain gauge adjustment, but over the years the product generation was continuously further developed and optimized. Next to quality control and correction of radar artefacts, the gauge adjustment changed. However, innovations always bring with them a discontinuity in the series of measurements. For this reason, processing of the RADOLAN data for climatological questions was started in June 2014 as part of the "Radar Climatology" project.*

→ *Quasi gauge-adjusted five-minute precipitation rate (YW): Winterrath, Tanja; Brendel, Christoph, Hafer, Mario; Junghänel, Thomas; Klameth, Anna; Lengfeld, Katharina; Walawender, Ewelina; Weigl, Elmar; Becker, Andreas (2018): RADKLIM Version 2017.002: Reprocessed quasi gauge-adjusted radar data, 5-minute precipitation sums (YW) DOI: 10.5676/DWD/RADKLIM_YW_V2017.002*

Lines 160 – 161 "an extended rain gauge adjustment with supplementary local rain gauges": How many rain gauges were used, what time step was used and what kind of adjustment have the authors applied?

→ *In order to ensure a comparable standard, we stuck to the same methodology for the rain gauge adjustment that the original RADOLAN/RADKLIM data is based on, but just with the additional rain gauges (Annotation: the original RADOLAN/RADKLIM product is already rain gauge adjusted → see RADKLIM and RADOLAN documentation stated at the previous reply). Thus, a densification of the measuring network is (in comparison to the original product) achieved. The adjustment technique is the best combination of the multiplicative and the additive adjustment (Bartels et al. (2004), Wood et al. (2000) and Wilson and Brandes (1979)). The time step used was*

*1 hour. From the radar perspective it is the 12 five minutes rain rates within one hour, which is the same time step used in the original RADOLAN/RADKLIM product.*

→ *The extra stations used, were – in Luxembourg - mainly the stations of the ASTA network (Administration des services techniques de l'agriculture) (ranging from 7 to 40 extra stations), and – in Germany – the stations of the agricultural-meteorological network of the state of Rhineland-Palatinate (ranging from 10 to 50 extra stations), which were quality controlled based on Sveruk, 1985 and Michaelides, 2008.*

→ *Bartels, H., Weigl., E., Reich, T., Lang, P., Wagner, A., Kohler, O. und Gerlach, N., (2004): Projekt Radolan. Zusammenfassender Abschlussbericht für die Projektlaufzeit 1997 bis 2004. https://www.dwd.de/DE/leistungen/radolan/radolan_info/abschlussbericht_pdf.pdf?__blob=publicationFile&v=2*

→ *Wood, S. J., D. A. Jones, and R. J. Moore (2000): Static and dynamic calibration of radar data for hydrological use, Hydrology and Earth System Sciences, 4(4), 545-554.*

→ *Wilson, J. W., and E. A. Brandes (1979), Radar measurement of rainfall - summary, Bull. Amer. Meteorol. Soc., 60(9), 1048-1058.*

→ *Sevruk, B. (1985). Correction of precipitation measurements summary report. In Correction of precipitation measurements. Swiss Federal Institute of Technology.*

→ *Michaelides, S. C. (Ed.). (2008). Precipitation: Advances in measurement, estimation, and prediction. Springer Science & Business Media.*

Lines 161 – 162 "We extracted the events for the database from the radar database by identifying 1x1 km grid cells with precipitation amounts ≥ 40 mm h$^{-1}$": But you do not have RADOLAN coverage in the full study area? Or is the study area constrained to the area covered by the RADOLAN observations?

→ *Unfortunately, the south-western part of the study area is not covered by the RADOLAN radar data. The ERA5 data was used from the entire squared study area. Flash floods were collected all over the study area, however, the included French regions are partly less densely populated and might under sample a bit. We will make this clearer when describing the database.*

Lines 172 – 174: Is this database giving all the floods for the study domain and which catchments does it contain?

→ *This sentence seems misleading now and we should delete "all". The question about catchments is a difficult one, as some floods occurred on hillslopes or streets that are within a catchment, but not really linked to its stream. In the database in the supplement, streams are mentioned, where they could be connected to the event.*

Lines 177 – 178: "The maximum hourly precipitation value was considered the trigger for the flash flood and atmospheric parameters were extracted from the identified grid cell and time.": What about the cells around this grid cell, as their parameters may also have influenced the rainfall that fell there?

→ *Averaging precipitation would lower the actual observed intensities, that are relevant for e.g. infiltration excess. Regarding atmospheric data, ERA5 is much coarser (0.25°x0.25°) than the radar dataset (1km x 1 km) and many thunderstorm cells. So, the atmospheric value that is extracted from ERA5 should be the same for neighbouring RADOLAN grid cells most of the time. Regarding neighbouring ERA5 grid cells, we have had some internal discussions. We discussed searching maxima/minima from the neighbouring ERA5 grid cells, as well as calculating the mean of 9 cells, to get a more representative value. This approach would however complicate the combined parameter analysis, as the mean values are sometimes higher/lower (especially for CAPE or CIN) than the values in one grid cell. We assumed that the large number of precipitation events statistically averages the differences.*

Lines 178 – 180: How did you find the flash floods here and the rainfall intensities, as this is outside the RADOLAN data coverage? In addition, do you have time series of the catchments, which could already indicate the presence and timing of a flash flood?

→ *There are only a few flash floods (11/83) outside the spatial (5) and the temporal coverage (6) of the RADOLAN data. For these we do not know the exact occurrence time and rainfall intensity. Unfortunately, we do not have discharge time series of these events.*

Line 203 "extremely rare in Central Europe": just out of interest (and perhaps worth mentioning), how rare is it (quantified)?

→ *This is an interesting question. Please find below the distribution of the K-Index within the study area & time. Out of 32235840 values in the grid cells of the study area, a K-Index $\geq 35°C$ occurred 99781 times, which equals 0.31 % of the cases. We will add "(< 0.5%)" to this sentence.*

[Figure]

*Figure 2: The distribution of the occurrence of the K-Index within the study area and time.*

Line 206 "700 hPa": Why have the authors chosen to pick the 700 hPa level?

→ *700 hPa is the middle of the lower, weather relevant part of the atmosphere between the surface and 500 hPa. This pressure level is a standard synoptic proxy used for simple, quick severe weather forecasts. The explanation for this somehow got lost in the final manuscript and will be added again.*

Line 215 "soil moisture (Swvl) $[m^3\ m^{-3}]$ at depths of 0-7 cm, 7-28 cm, and 28-100 cm from ERA5": Why have the authors chosen for these three depths and would it make sense to average them in some way, as they will be (cor)related to each other?

→ *These are the levels given by the ERA5 model. The fourth, deeper level (100-289 cm) was neglected as less relevant for fast runoff reactions and less sensitive. As they don't occur in any combined analyses, we think, that it is not important that they are related to one another. As especially the highest soil level seems to be of importance, averaging would straighten out this result.*

Lines 222 – 223 "Therefore, we chose upper or lower boundaries including 75 % of extreme events.": Do the authors mean the events IQR of the extreme events or did I understand it incorrectly?

→ *Depending on the parameters, we used the quartile as an upper or lower threshold but considered all events above or below. E.g. for the K-Index: "The higher, the more heavy precipitation events." Therefore, we selected all values above the lower quartile. We will revise the text to be clearer.*

Line 248 "Between 2001 and 2020, we observed a slight increase in the number of events per year (Figure 3a).": But not a significant one, right?

*→ Yes, that is true. We will be more precise with the wording.*

Lines 266 – 267 "Moisture conditions during extreme precipitation and flash flood events were found to be mostly within the upper percentiles of the overall simulated values.": That is also what you expect seeing the Clausius-Clapeyron (CC) relation and in fact even the 2CC relation for extreme precipitation. It probably deserves mentioning that, including some references (e.g. Lenderink & Van Meijgaard, 2008; Mishra et al., 2012; Manola et al., 2018; Wasko et al., 2018; Dahm et al., 2019).

*→ Yes, that is true. We will add this to the discussion.*

Lines 269 – 270 "All moisture parameters, and especially RH tend to be even higher during flash flood events compared to general extreme precipitation events (Figure 4d-f).": As clearly not all heavy rainfall events lead to flash flood events, can you also give some event statistics (earlier in the manuscript) between the two groups? What were average rainfall intensities in both groups, does the duration differ, does the size of the rainfall storms differ, etc.? This will give an idea why we see differences between the two groups. Lines 274 – 277: This also says a lot about the initial catchment wetness prior to a flash flood event. As stated earlier by the authors, the wetter, the quicker a flash flood can occur. Now, from these results, I do not directly see a significant difference between the three groups. Only the 'P' group has somewhat lower initial soil moisture values, which gives the impression that heavy, convective rainfall does more often occur during drier periods. Something which corresponds a bit to the summer weather patterns in Northwest Europe. It also suggests that initial soil moisture conditions were on average not different from other days in the studied periods, so the flash floods are mostly a result of the weather system and not initial conditions here.
In addition, perhaps it is interesting to show the soil moisture as a relative scale (so % of the capacity).

*→ Okay, we see the possible added value this comparison of P events that do lead to flooding and P events, where no flooding is reported in our database. We will try to add a short section about this to the manuscript. While it is not helpful to answer any of the hypotheses, it might give interesting additional information about the identification of atmospheric parameters and P events. However, regarding the large amount of P events in comparison to P events leading to flooding we are unsure, whether this brings clear results.*

*→ We should reformulate the text to be clearer. As we are comparing only soil moisture during P events and during FF events, there is a difference. We did not mention that normal conditions and the ones during FF are very similar. This indeed means that it is often dry before heavy precipitation events in the summer. However, if it is normal instead, then flooding is likelier.*

*→ The soil moisture in $m^3/m^3$ is on a relative scale already. Percentage of the capacity might indeed be a nice feature to better assess the saturation state of the soil, but not easily available. While volumetric saturation of the soil moisture, saturation of soil moisture and field capacity are partly available as an ERA5 dataset, we don't think this data makes real sense on such a course resolution. We dragged it along as a nice additional feature, but do not want to really put an emphasis on these results.*

Line 291 "These findings were particularly significant for the northern part of the study area (Figure 5b).": Any idea why in the north?

*→ No, unfortunately not. As CAPE is very sensitive to near surface moisture and temperature, this might be linked to a stronger increase in near-surface moisture. There might also be an orographic dependence that we could check.*

Lines 332 – 335: How is the trend if you take out 2016 and 2018?

→ *When setting 2016 and 2018 to 0 flash flood occurrences, the trend is a bit less strong (decrease in slope from 0.203 to 0.07), of course. Yet, it is still significant at p=0.005.*

[Figure]

*Figure 3: Occurrence of flash flood events within central western Europe between 1981 and 2020 as shown in the manuscript. Panel (a) shows the number of flash flood occurrences per year, panel (b) shows the same graph and trend analysis excluding 2016 and 2018.*

Lines 361 – 362 "Regarding low wind speeds and weak bulk shear, we found slightly increasing but barely significant trends.": But you did find a significant trend for LLS, right?

→ *Yes, we will rephrase it more precisely.*

Lines 380 – 382: This might also be related to the finite gauge-adjusted radar dataset of 20 years.

→ *"While atmospheric conditions tend to become more unstable, and overall warmer air masses potentially possess a higher amount of water vapour, the expected increase in (convective) precipitation events were not obvious from the analysed data." – Yes, this may be likely. We will add this and clarify, that the lengths of the time series differ.*

Lines 407 – 409 "Future analyses could incorporate the intra-annual temporal distribution of extreme precipitation events. Perhaps, formerly evenly distributed rainfall events tend to occur more condensed within a few days.": This is something you could already focus on in this study, by also looking at longer event durations. So what if you don't only look at 1-h accumulations, but also 6-h, 24-h, etc?

*→ This statement was referring to the clustering of events, e.g. like in 2016 or 2018, that might also be followed by a dry summer, but caused high casualties because of their accumulated occurrence within 1-2 weeks. I think it would require more than just looking at 6h or 24h accumulations, but also the accumulation of these on consecutive days and their accumulation before flash flood occurrences. While there are options to do these analyses with the dataset, it would require an extra supplement.*

Lines 420 – 422 "In addition to the hypothesis, we found mostly higher upper (0-7 cm) and lower (28-100 cm) layer soil moisture during flash flood events compared to general extreme precipitation events.": This did not seem that significant in the results.

*→ Compared to the precipitation events we do see (significant?) differences - compared to the overall values not. As we agreed on earlier, this says more about the pre-conditions during heavy precipitation events. It seems, that thunderstorms usually follow some sort of dry period. Yet, if they occur in wet or "normal" conditions, this might lead to a flash flood.*

Figure 1c: I suggest to put here an actual DEM with a higher resolution, which makes the mountain ranges and the differences between them clearer.

*→ As this Figure was only used for the explanation of model data, we opted to show the model topography. An actual DEM will be added as Fig. 1d. It will furthermore highlight the contrast in resolution.*

Figure 4: Would the differences (which are clearly present!) become clearer when you take the P and FF events out of the all group?

*→ We tried this nice suggestion but did not get clearer results. Using the current methods there are only 6588 P events and 84 FF events (out of which 45 are overlapping) that compare to a total of 32235840 values in the "all" section. Therefore, the elimination of these few data points did not make a visible difference.*

Figure 5:
1. An idea for the figure, make the colour scale discrete instead of continuous, then it is easier to distinguish the actual values.

*→ As with simulated values we are untrue about the "truth" of the "actual values". Therefore, we prefer the continuous scale, as we believe that it gives good enough tendencies and directions of the values.*

2. In addition, the slope is in [unit] per year. So, don't forget to give the unit.

*→ Thanks. We will correct this.*

3. To get an idea of the timeseries underneath, could the authors provide for one pixel the timeseries + trend?

*→ This can be added to the supplements. We can try to find a somewhat representative pixel, if that exists for the varying trends. In the conference contribution below, there are the timelines of the grid cell in Eastern Luxemburg, where some flash floods occurred in 2016 and 2018. (Meyer, J., Douinot, A., Neuper, M., Mathias, L., Tamez-Meléndez, C., Zehe, E., and Pfister, L.: Identifying and linking flash flood prone atmospheric conditions to flooding occurrences in central Western Europe, EGU General Assembly 2021, online, 19–30 Apr 2021, EGU21-12522, https://doi.org/10.5194/egusphere-egu21-12522, 2021).*

**Technical corrections**

→  *Thank you for pointing out these errors. We will adjust them accordingly.*

---

## Author Comment (AC3)

*Authors' response to Reviewer 3*

*[hess-2021-628-RC2]*

We thank the reviewer for his evaluation of our manuscript and his many helpful comments (hess-2021-628). Below we address the reviewer's comments (full text) indented by arrows and coloured in blue. We appreciate the efforts by the reviewer, which will help to improve our manuscript.

**General comments**

A major limitation of this study is the lack of flash flood events, particularly before 2006, and how they are identified. While the authors acknowledge this limitation in lines 332-333, I wonder if this is not an issue of a lack of flash flood events in the past, but a limitation of the observational record they use to define flash flood events. Flash floods are defined by news reports, prior literature (I am guessing case studies?), water agency reports, and reinsurance data, which are all prone to human error, including the need for people to observe the flood and report it as noteworthy. I wonder if they can incorporate any physically based indications of a flood event by including streamflow observations. This could address the dearth of floods prior to 2006 and remove some of the inherent bias in the human-based indications of flooding.

> → *As mentioned in detail in the responses to RC1 and RC2, we had collected a discharge database for the entire Moselle catchment, before starting the collection of data through newspapers, reports, etc.. From this data, it was impossible to clearly isolate flash floods in a statistic way. Moreover, the data is not consistent in time either, as long time series are available mainly for large rivers, but not the ones, in which flash floods could occur. Many of the now counted events also occurred in areas without measurements. Therefore, as cleanest option, we offer to take back the sub hypothesis 1 about the trend in flash floods and focus on the atmospheric background during the identified events.*

The connection between the flash flood events and extreme rainfall is unclear in the present form of the manuscript. The only description of how these events is linked are in lines 172–173 where the authors state that they include all floods that directly follow extreme precipitation. What is the temporal scale used to determine what "directly follows" means? Is this an hour, several hours, or a day? Please be explicit in stating this. Also, what is the spatial requirement for a flood event being connected to an extreme precipitation event? Please describe this in more detail. Finally, the independence of flood and precipitation events must be discussed. For example, if a flood event occurs on two consecutive days, is that considered the same event? Again, please discuss this in more detail.

> → *While it is mostly impossible to quantify the exact lag time for most events, as we don't know the time of the flood peak, "within a couple of hours" should cover all events. We will be more precise with that.*
> → *Regarding the spatial requirement: In order to connect a precipitation event and a flash flood event, they have to occur both within the same RADOLAN grid cell (1km\*1km) for now. We will however adjust this, as about half of the flash floods failed to be connected to a precipitation event. We will state the new method of matching events precisely in the method section. We will check if a e.g. 5 km radius can already be sufficient to identify related rainfall events.*
> → *For now, precipitation events are grids that cross the 40 mm/h threshold. These are accumulated to the size of ERA 5 grids. If it is crossed during 3h in a row it is for now counted as 3 rainfall events. This will be adjusted for the revised version. We will calculate events more precisely in their temporal and spatial resolution to make sure that one event matches a real*

*event. Regarding the further approach of identifying the atmospheric conditions during P events, we will stick to the approach of extracting atmospheric parameters of every ERA5 grid in which at least one RADOLAN grid cell has exceeded the threshold.*

*→ As we didn't use discharge data for floods, but reports only, we didn't count an overnight event twice. In one situation in Luxemburg, two floods occurred within two weeks. These were however two independent events (except for the elevated soil moisture and meso-scale atmospheric condition). If two floods occur in neighbouring catchments, they were also counted as two events*

It seems odd to me that despite an increase in more favorable rainfall environments over time, little trend is observed in changes to extreme rainfall. While the authors discuss possible reasons why this is in the discussion section, I wonder if it would be helpful to include other sources of rainfall data (like rain gauges or even ERA5, despite their limitations), to see if the same lack of a trend is reproduced.

*→ As also mentioned more in detail in the reply to RC2, we have analysed daily precipitation station data within the Moselle catchment in a previous approach (Meyer et al., 2020). As we could not identify trends in the stations, where long term data (since 1954) was available, we assumed, that the resolution was too coarse and we had missed too many thunderstorm events in the coarse network. Therefore, we choose to analyse the radar data. We do believe that the limitations of ERA5 data are indeed too high to be a valid additional source. The resolution is very coarse, such that thunderstorms and localized precipitation maxima are not properly simulated and thus significantly underestimated.*

*→ Meyer, J., Douinot, A., Zehe, E., Tamez-Meléndez, C., Francis, O., and Pfister, L.: Impact of Atmospheric Circulation on Flooding Occurrence and Type in Luxembourg (Central Western Europe), EGU General Assembly 2020, Online, 4–8 May 2020, EGU2020-13953, https://doi.org/10.5194/egusphere-egu2020-13953, 2020*

**Specific comments**

Lines 59–62: There are some contradictions in these lines as to how you refer to precipitation events that trigger floods. In line 59, you state that they are characterized by high rainfall amounts over a short period of time, while in the next few lines, you say that the rainfall also lasts over longer periods of time. What do you mean here? Please be clear if these are short or long duration rainfall events. Perhaps providing typical durations could be helpful here.

*→ This sounds indeed contradicting and we will reformulate this to be clearer. We are referring to time scales from 30 minutes to a few hours. The first "short" could be replaced by "sufficient" and the second, we can try to quantify to the above-mentioned time.*

Lines 67–70: I am not sure I understand what you are saying here–which processes are you referring to? Do you mean the upscale growth of convective cells into organized convection, like a mesoscale convection system? Please be more specific.

*→ Yes, upscale growth means that individual cells merge to form a mesoscale convective system (multicell storm). This can happen in an organised way along a well-defined boundary or in a more chaotic way (random cell clustering). We will revise the text to be clearer. Thank you for pointing this out.*

Line 80–81: I would also cite Schroeder et al. (2016) regarding a larger warm cloud depth leading to higher precipitation efficiency.

Schroeder, A., et al., 2016: Insights into atmospheric contributors to urban flash flooding across the United States using an analysis of rawinsonde data and associated calculated parameters. J. Appl. Met. and Clim., 55. Doi: https://doi.org/10.1175/JAMC-D-14-0232.1

→ *Thank you! We will add this source here.*

Lines 83–84: Large rainfall systems can also result in long duration storms (Doswell et al. 1996).

→ *We will rephrase more precisely and comprehensively.*

Line 88: I would start a new paragraph here at "Proxy parameters.." since this paragraph is already quite long.

→ *Thank you, we will consider restructuring this paragraph.*

Lines 91–93: The results of this study seem to contradict your previous sentence stating that bulk wind shear can be used to estimate precipitation efficiency if heavy precipitation occurs over a variety of DSL values. How do you reconcile this conflict?

→ *Weak DLS is present during weakly organized and slow-moving storms. In stronger DLS, cells are organized better and training/back-building multicell storms are possible. In our study area, the first case seems to be the dominant one. Nonetheless, this is a good point, and we will point out the precipitation efficiency and shear conflict a bit more as also already shown by Fankhauser (1988)*

→ *Fankhauser, J. C. (1988). Estimates of Thunderstorm Precipitation Efficiency from Field Measurements in CCOPE, Monthly Weather Review, 116(3), 663-684.*

Line 114: Please also cite Rasmussen et al. (2017), as they were among the first to discover the increasing CAPE/decreasing CIN paradigm:

Rasmussen, K. L., A. F. Prein, R. M. Rasmussen, K. Ikeda, and C. Liu, 2017: Changes in the convective population and thermodynamic environments in convection-permitting regional climate simulations over the United States. Climate Dyn., 55, 383–408, https://doi.org/10.1007/S00382-017-4000-7.

→ *Thank you for the reference, we will add it.*

Line 175–176: Are this 8 neighboring grid cells centered around the precipitation event grid cell? What if the precipitation event takes up multiple grid cells?

→ *This sentence as well as the approach are indeed confusing and will be revised. What we did, was picking the ERA5 grid cell at the location of the flash floods at the time the precipitation threshold of 40 mm/h was exceeded for the first time. As many flash floods were not connected to a P event with this method, we will improve our approach by looking at the neighbouring ERA5 grid cells as it is described in the manuscript. Generally, we had counted one precipitation event accumulating all 1*1km RADOLAN grid cells exceeding 40 mm/h to the margins of an ERA5 grid cell. For the sub-hypothesis 2, we will accumulate precipitation cells*

*exceeding thresholds better, independent of ERA5 cells, but dependant on their actual spatial and temporal resolution within the 1*1km grid width. This should however not impact the choice of ERA5 data, that is described here and will refer to the closest ERA5 grid cell in which the precipitation threshold is exceeded. From this grid cell, we picked the atmospheric data. If an event took up multiple grid cells, only the closest one was considered.*

Lines 205–215: Did you perform a sensitivity test to see if taking the RH or winds at different pressure levels aside from 700 hPa affected your results?

→ *This is a good point, we have added a supplement with analyses for the pressure levels of 500 hPa and 850 hPa. Looking at differing pressure levels gave a good overview over tendencies in the atmosphere. The positive trends in specific humidity (q) remain the same, are however less strong and less significant at 850 hPa compared to 700 hPa. At 500 hPa the original threshold of 0.004 kg kg$^{-1}$ is not crossed. We did not identify new thresholds depending on pressure levels for this supplement. While trends do differ in a few grid cells for RH, the overall conclusions drawn from the proxy level of 700 hPa remain valid, as results are not significant. Only for the absolute values of relative humidity (RH) above the threshold of 50% we do see a significant decrease at the lower atmospheric level of 850 hPa that is stronger than at higher levels of the atmosphere. For wind speed, we see trends, that are stronger negative at 500 hPa and therefore also for the mean between the surface level of 10 m and 500 hPa. These trends are however insignificant as well. At 850 hPa, decreases in wind speed are a bit more significant, but at a small range. As we concluded that there are no significant trends in the parameters selected regarding system motion and organisation, the conclusion can be kept also under consideration of the other values. The same stays true for the moisture parameters.*

Line 240: how would your results look if you omitted the year 2016? Would it still be an increasing trend?

→ *As graphically shown in the response to the second reviewer, a positive trend in the flash floods would remain when eliminating both, 2016 and 2018. Therefore, we assume, that it would also be the case when only omitting 2016.*

Line 258: what does "all hourly values" refer to? Is it the time of the precipitation event and the flood event combined or something else? This needs to be described in more detail. Also, how long, on average, are the precipitation events and the flood events?

→ *"All hourly values" are all values as available in the ERA5 dataset independent of P and FF events, so including P and FF events but also all other times without any event. We will rephrase this to be more precise.*

→ *The precipitation events are 1 hour long. If the threshold was crossed in 2 hours following each other, it is counted as two events. While this is not a natural event definition, we will reshape this for the second sub-hypothesis and also consider this for the ERA5 data. The flash flood events do not have a defined duration.*

Lines 261–262: Half of the distribution is above 100 J Kg$^{-1}$ of CIN, which is moderate CIN, so I do not believe saying high values of CAPE are often accompanied by low values of CIN is entirely accurate.

→ *We agree and will rephrase this sentence to "Sufficient values of CAPE are often accompanied by moderate values of CIN."*

Lines 273–274: Given the low wind speed and weak DSL, this likely indicates that these storms are slow-moving single-cell thunderstorms. This is interesting, because many flood-producing storms tend to be larger and more organized mesoscale convective systems (Ashley and Ashley 2008; Schumacher and Johnson 2006):

Ashley, S. T., and W. S. Ashley, 2008a: The storm morphology of deadly flooding events in the United States. Int. J. Climatol., 28, 493–503, https://doi.org/10.1002/joc.1554.

Schumacher, R. S., and R. H. Johnson, 2006: Characteristics of U.S. extreme rain events during 1999–2003. Wea. Forecasting, 21, 69–85, https://doi.org/10.1175/WAF900.1.

> → *Indeed, as also mentioned in one of the previous comments above. Thank you for pointing out this difference as well as referring us to the related references. We will add a paragraph about this in the discussion.*

Line 282: CAPE at or exceeding 100 J kg-1 is not high, but rather weak. I recommend you edit language throughout the paper to reflect this.

> → *This is true that CAPE of 100 J kg$^{-1}$ is not extremely high, but it is in the upper quartile of most values. We will consider revising this to "sufficient CAPE".*

**Technical corrections:**

> → *Thank you for your comments. We will change the text accordingly.*

---

## Author Comment (AC4)

*Supplement of*

**Increases in flash flood events and flash flood favouring atmospheric conditions in temperate regions of Europe**

Judith Meyer[1,2], Malte Neuper[3], Luca Mathias[4], Erwin Zehe[3], Laurent Pfister[1,2]

[1] Catchment and Ecohydrology Group (CAT), Environmental Research and Innovation, Luxembourg Institute of Science and Technology (LIST), Belvaux, 4422, Luxembourg
[2] Faculty of Science, Technology and Medicine (FSTM), University of Luxembourg, Esch-sur-Alzette, 4365, Luxembourg
[3] Institute of Water Resources and River Basin Management, Karlsruhe Institute of Technology (KIT), Karlsruhe, Germany
[4] Air Navigation Administration, MeteoLux, Findel, Luxembourg

*Correspondence to*: Judith Meyer (judith.meyer@list.lu)

**S2: Comparison of trends on differing pressure levels (500 hPa, 700 hPa, 850 hPa) for specific humidity (q), relative humidity (RH), and wind speed (WS). These figures are related to Figure 5 in the manuscript.**

**S2.1 Specific humidity (q) at 700 hPa and 850 hPa**

For this supplement we have used the same thresholds as initially identified for the 700 hPa level. As specific humidity is decreasing with height, we were not able to count enough values above 0.004 kg kg$^{-1}$ at 500 hPa to be able to plot or compare them. At 850 hPa, the positive trend found at 700 hPa remains (Figure 1 e, g). It is, however, less strong in the East of the study area, where it is moreover insignificant (Figure 1 f).

[Figure]

**Figure 1: Trend analysis of the specific humidity (q) above the identified threshold of 0.004 kg kg-150% at two differing pressure levels: 700 hPa and 850 hPa. The first column (a, e) shows the trends of the numbers of hourly occurrences of values above the threshold, including their significance-levels p in the second column (b, f). The third column (c, g) shows the trends of the mean values of all hourly occurrences above the threshold and the last column (d, h) their respective significance-levels p. White areas mark insignificance.**

[Figure]

**Figure 2: The difference of trends in specific humidity (q) above 0.004 kg kg$^{-1}$ yr$^{-1}$ between the pressure levels 850 hPa and 700 hPa regarding the annual number of occurrences (a) and the actual values (b).**

**S2.2 Relative humidity (RH) at 500 hPa, 700 hPa, 850 hPa**

The decrease in relative humidity (RH) is stronger at lower levels of the atmosphere (850 hPa) (Figure 3 i, k), where especially the mean of high RH is decreasing at a significant level (Figure 3 l). At the 500 hPa pressure level, the decrease in the number of occurrences is also stronger (Figure 3 a, b), than at 700 hPa. The actual values above the threshold increased however insignificantly at a very low rate (Figure 3 c, d). The 700 hPa level therefore shows a good proxy in the middle of the troposphere.

[Figure]

**Figure 3: Trend analysis of the relative humidity (RH) above the identified threshold of 50% at three differing pressure levels: 500 hPa, 700 hPa and 850 hPa. The first column (a, e, i) shows the trends of the numbers of hourly occurrences of values above the threshold, including their significance-levels p in the second column (b, f, j). The third column (c, g, k) shows the trends of the mean values of all hourly occurrences above the threshold and the last column (d, h, l) their respective significance-levels p. White areas mark insignificance.**

45

[Figure]

**Figure 4: The difference of trends in relative humidity (RH) above 50% between the pressure levels 500 hPa and 700 hPa (a, b), as well as 850 hPa and 700 hPa (c, d).**

**S2.3 Windspeed (WS) at 500 hPa, 700 hPa, 850 hPa, and mean between 10m & 500 hPa**

50  At 500 hPa trends in windspeed ≤ 7 m s$^{-1}$ are showing a stronger, but insignificant decrease compared to the 700 hPa level (Figure 5 a-d, Figure 6 a, b). The decreases in the mean WS below 7 m s$^{-1}$ at the pressure level 850 hPa are even partly significant (Figure 5 k, l) and occur more often (Figure 5 i-j). Another common calculation of WS is the mean between the surface level (10 m above ground) and 500 hPa. It shows a stronger but insignificant decrease in the number of occurrences of low WS, as well as its mean compared to the 700 hPa level (Figure 5 m-p).

55

[Figure]

Figure 5: Trend analysis of the wind speed (WS) above the identified threshold of 7 m s$^{-1}$ at three differing pressure levels (500 hPa, 700 hPa and 850 hPa) as well as considering the mean between the surface (10 m) and the pressure level at 500 hPa. The first column (a, e, I, m) shows the trends of the numbers of hourly occurrences of values above the threshold, including their significance-levels
60  p in the second column (b, f, j, n). The third column (c, g, k, o) shows the trends of the mean values of all hourly occurrences above the threshold and the last column (d, h, l, p) their respective significance-levels p. White areas mark insignificance.

[Figure]

**Figure 6: The difference of trends in wind speed (WS) below 7 m s$^{-1}$ between the pressure levels 500 hPa and 700 hPa (a, b), as well as 850 hPa and 700 hPa (c, d), and the mean of 10 m and 500 hPa and 700 hPa (e, f).**

---

## Referee Report (RR1)

Review of: **Atmospheric conditions favouring extreme precipitation and flash floods in temperate regions of Europe** by Judith Meyer et al.

Ruben Imhoff

Ruben.Imhoff@deltares.nl

May 21, 2022

**Summary**

I would like to thank the authors for their extensive responses to the received reviews and the considerable effort they have made to improve the manuscript. The manuscript has improved considerably, and I think the authors present a more balanced story that is well supported by their results now. Below, I have a few additional comments and suggestions, but these are rather minor.

**General comments**

*Flash flood database (lines 171 – 179)*

In the response to my earlier comment, the authors wrote: *"Before the choice of using a database that was collected through various sources, we analysed discharge data in the region (entire Moselle catchment). Therefore, we collected data for time series as long as possible. Long times series are, however, mainly available for large rivers, such as the Moselle or other bigger stream gauges, but not for catchments, in which flash floods occur. Moreover, data is often only available on a daily resolution. We have conducted several analyses of specific discharge using 79 stations within the region with catchments < 300 km² and found it hard to extract flash floods or high floods from these data. High flows in the past (1980s) were often caused by zonal precipitation in the Vosges mountains. Some regional flash floods that were of major importance and that we know well (i.e. Ernz Blanche 2016 & 2018), were to some extent detected by discharge data, but the overall time series are too short for any long-term analysis. Other events were so small and even outside streams, that they were not even captured by any stream gauge. We concluded that the inconsistencies in this type of streamflow-based dataset would be even bigger than the one presented in the manuscript. Apart from actual flash floods we have also made analyses about the number of scientific reports on the topic, which also started to increase around that time period (beginning 2000), when the topic received more attention. While a better database would be desirable, flash floods rely on site inspections."*

I think this information is actually very relevant for the reader. Can I ask the authors to put parts of their answer above in the text (either here or in the discussion section)?

**Specific comments**

Lines 146 – 147 "Unfortunately, the south-western part of the study area is not covered by the RADOLAN data": Perhaps add a reference to Figure 1b here.

Lines 294 – 296 "Often, soil moisture within the upper and lower soil layer (Swvl10-7 cm, Swvl37-100 cm) is higher during flash flood events compared to general extreme P events (Figure 295 5k, m). The mid-level soil layer (Swvl27-28 cm) shows lower soil moisture before flash flood events (Figure 5l).": These lines still lack some interpretation in my opinion, i.e. do you expect to see these differences between upper/lower and mid layers?

Lines 301 – 303 "Moreover, sufficient CAPE, high q and weak WS10m-500hPa were identified as the most clearly distinguishing parameters per category to characterize extreme precipitation events, including 75% of all extreme precipitation events and excluding around 75% of all generally occurring parameters values": What about the K-index? No need to change the top three parameters in my opinion, but good to mention the strong signal in this parameter (as the authors already do in their conclusion).

Lines 416 – 417 "In recent years they have been increasingly observed, especially in summer (Detring et al., 2021; Lupo, 2020)": You could also add a reference to the July 2021 floods here, for instance Kreienkamp et al. (2021).

Figure 5 and lines 291 – 293: I think I haven't mentioned this in my previous review, but I can imagine that the difference between the P and FF classes and non-extreme rainfall events might even be larger than the current comparison with "all" classes, as this also included many no-rain time steps (which may also have relatively low wind speeds and shear levels). It is just an idea, but perhaps worth the try if it makes the conclusions stronger.

**Technical corrections**

Line 35 ".Flash flood": Seems like you have forgotten a space between the dot and "flash".

Line 371 "Moselle catchment": for the non-European readers, perhaps briefly mention the location of this catchment.

Line 406 "US American studies": Just American studies would suffice.

Conclusion section: it may help readers that quickly scan through the paper, to write out the abbreviations once again in the conclusions.

Figure 1: The caption still only contains subfigures (a) – (c), while there are four subpanels now.

Figure 2: Very useful addition to the text!

**References**

Kreienkamp, F., Philip, S. Y., Tradowsky, J. S., Kew, S. F., Lorenz, P., Arrighi, J., Belleflamme, A., Bettmann, T., Caluwaerts, S., Chan, S. C., Ciavarella, A., De Cruz, L., de Vries, H., Demuth, N., Ferrone, A., Fischer, r. M., Fowler, H. J., Goergen, K., Heinrich, D., Henrichs, Y., Lenderink, G., Kaspar, F., Nilson, E., Otto, F. E. L., Ragone, F., Seneviratne, S. I., Singh, R. K., Skålevåg, A., Termonia, P., Thalheimer, L., van Aalst, M., Van den Bergh, J., Van de Vyver, H., Vannitsem, S., vanOldenborgh, G. J., Van Schaeybroeck, B., Vautard, R., Vonk, D. andWanders, N. (2021) Rapid attribution of heavy rainfall events leading to the severe flooding inWestern Europe during July 2021. WorldWeather Atribution. URL: http://hdl.handle.net/1854/LU-8732135.

---

## Referee Report (RR2)

Meyer et al. 2021, "More frequent flash flood events and extreme precipitation favouring atmospheric conditions in temperate regions of Europe"- Revised version

**General comments:**

The authors did a good job of addressing my previous comments and have greatly improved the clarity and soundness of their manuscript. In particular, the methods are explained more clearly now and the incorporation of additional data sources and analysis has made the results more robust. I think there are still a couple of remaining issues that need to be addressed before publication, namely the following:

1) Given the temporal inconsistencies in the flash flood reports and lack of a long enough record, I do not believe the linear trend analysis of flash flood occurrence (Fig. 3a) should be included or discussed in depth. The authors even state themselves that "the dataset do not allow drawing conclusions on any robust trends", so why include this figure given its potential to mislead readers? I think it is enough to just state that the linear trend analysis is inconclusive due to the data issues.

2) Additional clarity is needed to distinguish the precipitation events versus the subset that are associated with flash floods. I recommend something like a table to show the number of total precipitation events and the number of precipitation events that are associated with flash floods. That could help make the results more generalizable.

**Specific comments:**

- Lines 51-53: In the U.S., there are nice definitions of flash floods used by the National Weather Service- perhaps you can utilize that or something similar that exists in Europe? In the U.S., the NWS defines a flash flood as "a rapid and extreme flow of high water into a normally dry area, or rapid rise in a stream or creek above a predetermined flood level, beginning within six hours of the causative event" (NWS 2021.)

- Line 59-60: I do not believe this is entirely correct description of storm training- please revise this sentence to reflect that "echo training" is when convective cells move in the line-parallel direction leading to repeated cell motion over an area (Peters and Schumacher 2015):
    - Peters, J.M. and R.S. Schumacher, 2015: "Mechanisms for organization and echo training in a flash flood-producing mesoscale convective system". Mon. Wea. Rev., 143, 1058-1085. Doi: https://doi.org/10.1175/MWR-D-14-00070.1.

- Lines 62-64: Likewise, this sentence needs revising, as forward movement is not halted. Rather the direction of the cell motion and propagation vector cancel out leading to new cells being continuously generated over the same area (Doswell et al. 1996).

- Line 103: I recommend revising this sentence because the more intense thunderstorms are actually triggered because of high CAPE **and** high CIN.

- Fig. 1d: What is this panel? It is not labeled in the figure caption. Please either omit or add its description to the caption.

- Line 150: Do the supplementary rain gauges cover the same time period?

- Line 169-170: I think it would be helpful to at least briefly describe this procedure in one sentence, like you do in the figure caption below.
- Line 175-176: How did you determine the spatial threshold of 30 km? Likewise, how did you determine the temporal threshold of one day? What happens if you have multiple hours of precipitation (which count as separate events according to your definition) and one flood?
- Figure 2: This is a very helpful figure, although I am a bit confused about the difference between the dashed and solid lines- is one dashed box an ERA5 grid cell and one solid box the multiple ERA5 grid cells used to take the atmospheric condition? If so, please make that clear in the figure description.
- Line 207: How did you obtain 0.5%? Did you calculate it yourself or did you find it in the literature?
- Line 263-264: it is difficult to tell from Figure. 4, but to me it looks like the median line for max hourly intensity is actually higher in flash flood events than for all P events- can you please see if this is true and provide numbers for these values?
- Figure 4a: I believe the text that states "P events associated with flash floods" is incorrect here, because the text states that these are all P events in the summer- is that true?
- Line 294-296: This is a very interesting result!
- Table 2: Is this table for all extreme P events or just those that are associated with flash floods? It would be interesting to show the values for both events.
- Line 389: Results are either significant or not- please pick one.
- Line 389-390: I don't believe you can state that the storm organization is unchanged, as you did not explicitly study changes in storm structure.
- Line 391: Future studies actually show a decrease in shear with warming and it would be helpful to cite those studies here (Diffenbaugh et al. 2013, Brooks 2013).
  - Brooks, H.E., 2013: Severe thunderstorms and climate change. *Atmospheric Research*, **123**, 129-138. https://doi.org/10.1016/j.atmosres.2012.04.002.
  - Diffenbaugh, N.S., M. Schere, and R.J. Trapp, 2013: Robust increases in severe thunderstorm environments in response to greenhouse forcing. *PNAS*, **110**, 16361–16366, https://doi.org/10.1073/pnas.1307758110.

---

## Author Response (AR2)

**Authors' response to Editor decision [hess-2021-628]**

*We thank the editor for the opportunity to revise the manuscript and addressed the reviewers' additional comments as indicated below.*

*In this version we have:*

- *reformulated all conclusions about an actual increase in flash floods due to data scarcity*
- *clarified figures and legends (Fig.2, 3 & 4)*
- *made some textual changes based on comments.*

**Authors' response to Reviewer 1 [hess-2021-628-RC1]**

*We thank the reviewer Ruben Imhoff for his evaluation of our manuscript and his many helpful comments (hess-2021-628). Below we address the reviewer's comments (full text) indented by arrows and coloured in blue. We appreciate the efforts by the reviewer to take another look at our manuscript and give ideas to further improve it.*

**General comments**

**Flash flood database (lines 171 – 179)**

In the response to my earlier comment, the authors wrote: *"Before the choice of using a database that was collected through various sources, we analysed discharge data in the region (entire Moselle catchment). Therefore, we collected data for time series as long as possible. Long times series are, however, mainly available for large rivers, such as the Moselle or other bigger stream gauges, but not for catchments, in which flash floods occur. Moreover, data is often only available on a daily resolution. We have conducted several analyses of specific discharge using 79 stations within the region with catchments < 300 km² and found it hard to extract flash floods or high floods from these data. High flows in the past (1980s) were often caused by zonal precipitation in the Vosges mountains. Some regional flash floods that were of major importance and that we know well (i.e. Ernz Blanche 2016 & 2018), were to some extent detected by discharge data, but the overall time series are too short for any long-term analysis. Other events were so small and even outside streams, that they were not even captured by any stream gauge. We concluded that the inconsistencies in this type of streamflow-based dataset would be even bigger than the one presented in the manuscript. Apart from actual flash floods we have also made analyses about the number of scientific reports on the topic, which also started to increase around that time period (beginning 2000), when the topic received more attention. While a better database would be desirable, flash floods rely on site inspections."*

I think this information is actually very relevant for the reader. Can I ask the authors to put parts of their answer above in the text (either here or in the discussion section)?

**Specific comments**

Lines 146 – 147 "Unfortunately, the south-western part of the study area is not covered by the RADOLAN data": Perhaps add a reference to Figure 1b here.
→ *Good suggestion, we will add the reference.*

Lines 294 – 296 "Often, soil moisture within the upper and lower soil layer (Swvl10-7 cm, Swvl37-100 cm) is higher during flash flood events compared to general extreme P events (Figure 295 5k, m). The mid-level soil layer (Swvl27-28 cm) shows lower soil moisture before flash flood events (Figure 5l).": These lines still lack some interpretation in my opinion, i.e. do you expect to see these differences between upper/lower and mid layers?

→ *We can extend the speculation in the discussion section a bit (lines 420-427) that the soil is often dry during the summer, but that preceding rainfall events wettened the top layer that eventually hints towards quicker runoff generation in terms of infiltration excess overland flow. We had expected to see wetter top layers and didn't have expectations about lower layers.*

→ *Overall, we however prefer to keep the focus on the atmospheric conditions and not invest too much into soil moisture.*

Lines 301 – 303 "Moreover, sufficient CAPE, high q and weak WS10m-500hPa were identified as the most clearly distinguishing parameters per category to characterize extreme precipitation events, including 75% of all extreme precipitation events and excluding around 75% of all generally occurring parameters values": What about the K-index? No need to change the top three parameters in my opinion, but good to mention the strong signal in this parameter (as the authors already do in their conclusion).

→ *Yes, true, it seems to have gotten lost from your last comments. We will add a small interpretation of the K-Index to chapter 3.4, as this comment especially refers to the strong trends of the K-Index. We did not do that so far, as it is an index and not a more or less independent parameter. Yet, CAPE is also calculated, so we can add the interpretation of the K-Index accordingly.*

Lines 416 – 417 "In recent years they have been increasingly observed, especially in summer (Detring et al., 2021; Lupo, 2020)": You could also add a reference to the July 2021 floods here, for instance Kreienkamp et al. (2021).

→ *While the July 2021 flood mechanism differs a bit from the other, more or less isolated flash floods that we are considering, we agree, that they also occurred during an atmospheric blocking situation and are subject to events within recent years. We will add a corresponding reference here.*

Figure 5 and lines 291 – 293: I think I haven't mentioned this in my previous review, but I can imagine that the difference between the P and FF classes and non-extreme rainfall events might even be larger than the current comparison with "all" classes, as this also included many no-rain time steps (which may also have relatively low wind speeds and shear levels). It is just an idea, but perhaps worth the try if it makes the conclusions stronger.

→ *This is a good idea to split the data into more subsets. However, the rain radar data only starts in 2002, while all ERA5 data is available from 1982. The original idea of including the "all" values, was to give a general idea of what values are observed in the study area throughout the entire time period.*

→ *Identifying light rainfall events is moreover tricky. To identify them correctly, we would have to look for events and merge cells, build averages etc., as we did for the heavy rainfall events. It would moreover be difficult to exclude the 'borders' with lighter rain of the heavy rainfall events.*

**Technical corrections**

→ *Thank you for your suggestions and pointing out these errors. We will adjust the manuscript accordingly.*

**Authors' response to Reviewer 2 [hess-2021-628-RC2]**

*We thank the reviewer for his evaluation of our manuscript and his many helpful comments (hess-2021-628). Below we address the reviewer's comments (full text) indented by arrows and coloured in blue. We appreciate the efforts by the reviewer, which will help to improve our manuscript.*

**General comments**

Given the temporal inconsistencies in the flash flood reports and lack of a long enough record, I do not believe the linear trend analysis of flash flood occurrence (Fig. 3a) should be included or discussed in depth. The authors even state themselves that "the dataset do not allow drawing conclusions on any robust trends", so why include this figure given its potential to mislead readers? I think it is enough to just state that the linear trend analysis is inconclusive due to the data issues.

→ *We will remove the light-grey dotted trend line and the trend values in Figure 3 (a) and an according sentence within the results description. We will also go through the text again and look for necessary adjustments.*

Additional clarity is needed to distinguish the precipitation events versus the subset that are associated with flash floods. I recommend something like a table to show the number of total precipitation events and the number of precipitation events that are associated with flash floods. That could help make the results more generalizable.

→ *Thank you for your suggestion. This would give a tiny table, that could look like this:*

|  | P events | P events associated with FF |
|---|---|---|

| No. of events | 3835 | 37 |
| --- | --- | --- |

→ *We will add the values to the manuscript.*

**Specific comments**

Lines 51-53: In the U.S., there are nice definitions of flash floods used by the National Weather Service- perhaps you can utilize that or something similar that exists in Europe? In the U.S., the NWS defines a flash flood as "a rapid and extreme flow of high water into a normally dry area, or rapid rise in a stream or creek above a predetermined flood level, beginning within six hours of the causative event" (NWS 2021.)

→ *This is an interesting and wide enough definition we had not yet come across. While we consider much smaller dimensions of floods in central Western Europe compared to the US and the Mediterranean, the character is the same as described in the definition of the NWS glossary https://forecast.weather.gov/glossary.php.*

Line 59-60: I do not believe this is entirely correct description of storm training- please revise this sentence to reflect that "echo training" is when convective cells move in the line-parallel direction leading to repeated cell motion over an area (Peters and Schumacher 2015): Peters, J.M. and R.S. Schumacher, 2015: "Mechanisms for organization and echo training in a flash flood-producing mesoscale convective system". Mon. Wea. Rev., 143, 1058-1085. Doi: https://doi.org/10.1175/MWR-D-14-00070.1.

→ *Thank you for your comment. We will revise this sentence in the manuscript.*

Lines 62-64: Likewise, this sentence needs revising, as forward movement is not halted. Rather the direction of the cell motion and propagation vector cancel out leading to new cells being continuously generated over the same area (Doswell et al. 1996).

→ *Thank you for your comment. We will revise this sentence in the manuscript.*

Line 103: I recommend revising this sentence because the more intense thunderstorms are actually triggered because of high CAPE **and** high CIN.

→ *Thank you. High CIN is often connected to isolated thunderstorm cells rather than large scale thunderstorm conditions, that would potentially lead to the rainfall amounts required for flooding. High CIN mainly indicates that strong lifting mechanisms are needed. High CAPE, however, always has the potential for intense thunderstorms due to strong possible updrafts of a cell.*

→ *We will revise this to "higher CIN levels may lead to higher CAPE values since it prevents premature initiation of convection potentially inhibiting the development of stronger CAPE, and thus possibly increasing the potential of more intense storms."*

Fig. 1d: What is this panel? It is not labeled in the figure caption. Please either omit or add its description to the caption.

→ *Sorry, we forgot to adjust the caption after adding the fourth panel. (d) is what is written for (c). (c) was added new and is an actual digital elevation model of the area at a 1x1 km resolution. We will adjust the caption accordingly.*

Line 150: Do the supplementary rain gauges cover the same time period?

→ *Not all rain gauges cover the same time period as the network became denser over the last years. As we wrote in the lines 155-158, the number of extra rain gauges ranged from 7 to 40 extra rain gauge stations in Luxembourg and 10 to 50 extra rain gauge stations in Germany. Of course, there is an inconsistency but that is always the case when using rain (gauge) data. The physical rain gauges and the environment (wind field, vegetation) around the stations change for example. Nevertheless, the rain gauges are just one part in the processing chain of data adjustment. The main data source is the radar data. Although the radar hardware also changes, as well as the basic radar (internal) quality control and correction algorithms. Thus, a constant time series is an unreachable ideal conception, but we used all sources we could to adjust data as close to reality as we could.*

Line 169-170: I think it would be helpful to at least briefly describe this procedure in one sentence, like you do in the figure caption below.

→ *Ok, we suggest to add: For a small standard P event that lies within one ERA5 grid cell, atmospheric data was averaged over that particular ERA5 grid cell and the eight surrounding ones. Precipitation events at the boundary of the study area do not include the full buffer zone and larger P events covering multiple grid cells include a buffer zone around the ERA5 grid cells of the actual P event.*

→ *We will also add to line 163: "For every P event, we extracted the maximum hourly precipitation intensity at one location within the P event as well as the maximum 5-minute precipitation intensity at one location within the P event."*

  o *5 minute max: within one grid cell at one time. Not averaged in space over the entire size of the P event*

  o *1 hour sum: moving window over time within one grid cell. Not averaged in space, only at one location*

Line 175-176: How did you determine the spatial threshold of 30 km? Likewise, how did you determine the temporal threshold of one day? What happens if you have multiple hours of precipitation (which count as separate events according to your definition) and one flood?

→ *The 30 km are to reach the next ERA5 grid cell in case no P event > 40 mm/h was identified in the one where the flood occurred. At the actual flooding location, which is not a point either, the hourly precipitation intensity might have been just below the determined threshold or the flood occurred a bit downstream of the P event. If a flood was triggered by a rainfall event not identified as extreme in the radar data, the flood was not considered.*

→ *There would have to be a major temporal and spatial gap (> ½ h, > 2 grid cells = 2 km) in the intense precipitation to make them count as two events. In this case the first would "only" contribute to the pre-event moisture according to our definition and the one closest to the*

*flood event caused the flood. By looking at the few floods individually we did not find discrepancies regarding this.*

Figure 2: This is a very helpful figure, although I am a bit confused about the difference between the dashed and solid lines- is one dashed box an ERA5 grid cell and one solid box the multiple ERA5 grid cells used to take the atmospheric condition? If so, please make that clear in the figure description.

→ *Yes, that is what it is. We will improve the legend and make this clearer in the figure description.*

Line 207: How did you obtain 0.5%? Did you calculate it yourself or did you find it in the literature?

→ *We calculated this value ourselves based on the ERA5 data of the area. We will state this clearly.*

Line 263-264: it is difficult to tell from Figure 4, but to me it looks like the median line for max hourly intensity is actually higher in flash flood events than for all P events- can you please see if this is true and provide numbers for these values?

→ *It is true that the median differs from 46.54 mm/h as the mean of all P events and 49.65 mm/h as the mean of all P events leading to flash floods. We will rephrase the sentence to: "P events that eventually led to flash floods (Figure 4c, e) do not differ in the range of precipitation intensities from P events that did not cause flash floods, but their median."*

Figure 4a: I believe the text that states "P events associated with flash floods" is incorrect here, because the text states that these are all P events in the summer- is that true?

→ *We will adjust the legend to be clearer. The blue crosses are indeed the P events associated with flash floods. These are however not shown in Figure 4 (a), just in the other panels. This line of the legend will be moved outside the plot panel.*

Line 294-296: This is a very interesting result!

→ *As indicated by the other reviewer, we add a few sentences of interpretation to the discussion section. Especially the higher moisture in the top soil layer hints to preceding rainfall events and might help explaining some of the quick runoff formation present during flash floods. We however prefer to keep the focus on atmospheric parameters and not go into too much detail regarding the soil moisture.*

Table 2: Is this table for all extreme P events or just those that are associated with flash floods? It would be interesting to show the values for both events.

→ *Table 2 refers to all extreme P events, independent of the occurrence of floods. We rate the threshold values of P events causing flooding, as less statistically robust, as only 37 events would contribute to their calculation. Moreover, they might confuse the reader as to which*

*thresholds were used. We have displayed the thresholds for both in the extended table below but prefer to not add the values to the manuscript.*

→ *We will update the table description to be clearer about the values to: "**Table 2:** Threshold values determined as extreme precipitation and flash flood favouring based on the lower/upper quartile of their range of occurrence during extreme precipitation events, including all P events, whether they are associated with a flood or not.*

| | Instability | | | Moisture | | | Storm motion & organisation | | | |
|---|---|---|---|---|---|---|---|---|---|---|
| | CAPE | CIN | Kx | TCWV | q | RH | $WS_{700\,hPa}$ | $WS_{10m\text{-}500hPa}$ | LLS | DLS |
| P | ≥ 326.9 $J\,kg^{-1}$ | ≤ 183.5 $J\,kg^{-1}$ | ≥ 27.8 °C | ≥ 26.5 $kg\,m^{-2}$ | ≥ 0.004 $kg\,kg^{-1}$ | ≥ 59.4 % | ≤ 7.1 $m\,s^{-1}$ | ≤ 6.2 $m\,s^{-1}$ | ≤ 3.8 $m\,s^{-1}$ | ≤ 10.4 $m\,s^{-1}$ |
| FF | ≥ 355.2 $J\,kg^{-1}$ | ≤ 126.4 $J\,kg^{-1}$ | ≥ 27.6 °C | ≥ 26.0 $kg\,m^{-2}$ | ≥ 0.004 $kg\,kg^{-1}$ | ≥ 63.4 % | ≤ 7.6 $m\,s^{-1}$ | ≤ 6.5 $m\,s^{-1}$ | ≤ 4.4 $m\,s^{-1}$ | ≤ 12.0 $m\,s^{-1}$ |

Line 389: Results are either significant or not- please pick one.

→ *We will rephrase the sentence to: "Increasing trends in low LLS are significant in the south-eastern part of the study area."*

Line 389-390: I don't believe you can state that the storm organization is unchanged, as you did not explicitly study changes in storm structure.

→ *Thank you for pointing this out. We will rephrase to: "Overall, the proxy parameters used for the assessment of organisation and motion of storm systems stayed largely unchanged with tendencies favouring the occurrence of extreme precipitation."*

Line 391: Future studies actually show a decrease in shear with warming and it would be helpful to cite those studies here (Diffenbaugh et al. 2013, Brooks 2013).
- Brooks, H.E., 2013: Severe thunderstorms and climate change. *Atmospheric Research*, **123**, 129-138. https://doi.org/10.1016/j.atmosres.2012.04.002.
- Diffenbaugh, N.S., M. Schere, and R.J. Trapp, 2013: Robust increases in severe thunderstorm environments in response to greenhouse forcing. *PNAS*, **110**, 16361–16366, https://doi.org/10.1073/pnas.1307758110.

→ *Thank you for these references. We will add them.*

---

## Author Response (AR3)

**Authors' response to Editor decision [hess-2021-628]**

*We thank the editor for the opportunity to revise the manuscript again and addressed the reviewers' additional comments as indicated below. Thanks to the reviewers for continuously improving our manuscript.*

*In this version we have:*

- *Extended the section in the discussion about the numbering of the P events*
- *clarified figures (Fig.4 d) and captions (Fig. 5)*
- *made some technical changes based on the reviewer's comments*

**Authors' response to Reviewer 3**

*We thank the reviewer for taking the time to review our manuscript and support us with further helpful comments (hess-2021-628). Below we address the reviewer's comments (full text) indented by arrows and coloured in blue.*

**General comments**

The authors sufficiently addressed my previous comments and I especially appreciate the work they put into to creating the supplementary figures. It is clear the authors put a lot of time into revising this manuscript and it shows. Overall, I think this a nice study that is close to being ready for publication.

However, I still do have one remaining issue, though this could be an issue of opinion. In their response about how they count multiple hours of precipitation as separate events, the authors responded, "There would have to be a major temporal and spatial gap (> 1⁄2 h, > 2 grid cells = 2 km) in the intense precipitation to make them count as two events. In this case the first would "only" contribute to the pre-event moisture according to our definition and the one closest to the flood event caused the flood. By looking at the few floods individually we did not find discrepancies regarding this."

I do not think ½ h is a major temporal gap, as storms like back-building thunderstorms or long duration storms like the Colorado floods of 2013 (Gochis et al. 2015) could result in precipitation gaps > ½ h. Therefore, I do not think it is fair to count these as separate events. Thus, the number of precipitation events might be inflated. Yet, only counting the precipitation event closest to the flood mitigates an artificially high number of precipitation induced floods. I do not think the authors have to change their method, but a note about this caveat in the discussion section would be helpful.

→ *In comparison to American or Mediterranean flash floods, events in temperate Europe occur on a much smaller scale. As indicated by the low DLS, most events occur during slow-moving*

*single cell thunderstorms. Back-building storms might not be sufficiently counted as a connected event, yet they also only present a minority of events. We agree that we might count more P events and that P events might appear too "small" to cause a flash flood, if it is not the P event itself leading to the flood, but a combination of storms. We will add this to the discussion.*

**Specific comments**

Line 95: Replace "focussing" with "focusing"
> → *Thank you.*

Line 126: I suggest replacing "apprehension" with "understanding" for greater clarity.
> → *Okay, we will revise this.*

Figure 4d: Can you change the y-axis of this figure so that the bottom tails of the boxplots can be seen? As it is currently plotted, the tails are not visible.
> → *The tails are all around 40 mm/h according to the lower limit of the selection method. We will adjust the axis length to make the tails visible.*

Figure 5 caption: How did you determine the time before flash flood events? Is it the start of the triggering P event you describe in the methods?
> → *Yes, it is the time respective to the onset of the P event for P and the triggering P event for FF. We will clarify this further in the method section, line 245, as well as in the mentioned caption.*

Line 435–437: Please add Dougherty and Rasmussen (2019) to the list of citations here:
   Dougherty, E. and K. L. Rasmussen, 2019: Climatology of flood-producing storms and their associated rainfall characteristics in the United States. Mon. Wea. Rev., 147, 3861–3877, doi: https://doi.org/10.1175/MWR-D-19-0020.1.
> → *Thank you for the reference. We will add it.*

Line 481: Replace "sheer" with "shear".
> → *Thank you.*